# Expression of CD226 is associated to but not required for NK cell education

Arnika K. Wagner[1,2,*], Nadir Kadri[2,*], Johanna Snäll[3], Petter Brodin[4,5], Susan Gilfillan[6], Marco Colonna[6], Günter Bernhardt[7], Petter Höglund[2], Klas Kärre[1,**] & Benedict J. Chambers[3,**]

DNAX accessory molecule-1 (DNAM-1, also known as CD226) is an activating receptor expressed on subsets of natural killer (NK) and T cells, interacts with its ligands CD155 or CD112, and has co-varied expression with inhibitory receptors. Since inhibitory receptors control NK-cell activation and are necessary for MHC-I-dependent education, we investigated whether DNAM-1 expression is also involved in NK-cell education. Here we show an MHC-I-dependent correlation between DNAM-1 expression and NK-cell education, and an association between DNAM-1 and NKG2A that occurs even in MHC class I deficient mice. DNAM-1 is expressed early during NK-cell development, precedes the expression of MHC-I-specific inhibitory receptors, and is modulated in an education-dependent fashion. $Cd226^{-/-}$ mice have missing self-responses and NK cells with a normal receptor repertoire. We propose a model in which NK-cell education prevents or delays downregulation of DNAM-1. This molecule endows educated NK cells with enhanced effector functions but is dispensable for education.

[1] Department of Microbiology, Tumor and Cell Biology, Karolinska Institutet, Nobels väg 16, 17177 Stockholm, Sweden. [2] Department of Medicine Huddinge, Center for Hematology and Regenerative Medicine, Karolinska Institutet, Hälsovägen 7, 14157 Huddinge, Sweden. [3] Department of Medicine Huddinge, Center for Infectious Medicine, Karolinska Institutet, F59, 14186 Stockholm, Sweden. [4] Science for Life Laboratory, Department of Medicine Solna, Karolinska Institutet, and Unit of Infectious Diseases, Karolinska University Hospital, 17176 Stockholm, Sweden. [5] Department of Neonatology, Karolinska University Hospital, 17176 Stockholm, Sweden. [6] Department of Pathology and Immunology, Campus Box 8118, Washington University School of Medicine, 660 South Euclid Avenue, St Louis, Missouri 63110, USA. [7] Institute of Immunology, Building 11, Hannover Medical School, Carl Neuberg Straße1, 30625 Hannover, Germany. * These authors contributed equally to this work. ** These authors jointly supervised this work. Correspondence and requests for materials should be addressed to B.J.C. (email: Benedict.Chambers@ki.se).

Natural killer (NK) cells are lymphoid cells of the innate immune system that can kill tumour cells or virus-infected cells[1–3]. NK-cell effector functions depend on the integrated input of signals from activating and inhibitory receptors[4]. Activating receptors recognize stress-induced molecules which induce phosphorylation events that may culminate in release of cytotoxic granules and cytokines[5]. Unstressed cells are protected from killing because of the expression of self-MHC class I (MHC-I) on their surface which act as ligands for dominant inhibitory receptors[6].

Inhibitory receptors, such as the Ly49 receptors (Ly49r) in mice or the killer-cell immunoglobulin-like receptors in humans bind to self-MHC-I molecules thereby ensuring tolerance towards healthy autologous cells. The CD94/NKG2A heterodimer recognizes HLA-E in human and Qa-1b in mouse[7,8]. Inhibitory receptors contain immunoreceptor tyrosine-based inhibitory motifs in their cytoplasmic tails which, upon receptor engagement, become phosphorylated and recruit protein tyrosine phosphatases such as the Src-homology 2 domain-containing phosphatases SHP-1 and SHP-2. SHP-1 and SHP-2, together with phosphatidylinositol 5-phosphatase 1 (SHIP-1), dephosphorylate key signalling molecules in NK-cell activation, thereby attenuating NK-cell responses[9–11].

NK cells express inhibitory Ly49r in a variegated fashion, such that each cell can have one receptor several different receptors, or no receptors[4,12]. As a consequence, not all NK cells are inhibited by interactions between inhibitory Ly49r and self-MHC-I molecules and these cells are therefore hypo-responsive to alterations in MHC-I expression. NK cells that express at least one Ly49r or NKG2A to interact with host MHC-I during development are educated to be hyper-responsive, enabling them to act when they encounter cells with downregulated MHC expression. The signalling events downstream of inhibitory NK-cell receptors leading to education remain uncharacterized; however, a requirement of SHP-1, SHIP-1 and inhibitory receptors with intact immunoreceptor tyrosine-based inhibitory motif for NK-cells education and the acquisition of a normal receptor repertoire has been shown[9–11].

There are several models to explain the host MHC-I-dependent education of NK cells[11,13,14]. In the licensing model, NK cells that express inhibitory receptors specific for self-MHC-I become responsive and gain the capacity to perform effector functions[11]. In the disarming model, NK cells are 'armed' with the capacity to kill and secrete cytokines by default, and the NK cells that fail to express self-specific inhibitory receptors will become hypo-responsive or 'disarmed'[13]. In the rheostat model, the amount of inhibitory input perceived by an individual NK cell calibrates the level of responsiveness by that NK cell continuously, and responsiveness can thus be tuned up as well as down[15]. Both the expression level of inhibitory receptors and the affinity of the NK-cell receptor for their MHC-I ligands determine the potential responsiveness of the NK cells[14–16].

DNAX accessory molecule 1 (DNAM-1, also known as CD226), a costimulatory adhesion molecule expressed by T cells and NK cells, has a crucial function in tumour immune surveillance[17–20]. The ligands of DNAM-1, CD155 and CD112, are expressed on many cell types including transformed or infected cells[18,20–24]. DNAM-1 associates with LFA-1, which increases adhesion to target cells. In human NK cells, the coordinated expression of DNAM-1 together with LFA-1 is linked to NK-cell education[25,26]. Furthermore, engagement of DNAM-1 alters the generation of MCMV-specific memory NK cells[27]. In addition, DNAM-1$^+$ NK cells in mice are more cytotoxic and produce higher levels of inflammatory cytokines than their DNAM-1$^-$ counterparts, when stimulated with IL-12 and IL-18 (refs 28,29). DNAM-1 is expressed on all human NK

cells; however, in mice, only about 60% of splenic NK cells express DNAM-1 (ref. 28).

DNAM-1 has been associated with specific inhibitory receptors, suggesting that it might be functionally involved in NK-cell education[25,28]. In the present study we show that DNAM-1 function is associated with NK-cell education but is not necessary to reach nor maintain the educated state.

## Results

**Reduced DNAM-1 expression in $B2m^{-/-}$ NK cells.** Since it has been suggested previously that there was a link between DNAM-1 and NK-cell education[25,28,30], we investigated the expression of DNAM-1 on NK cells from mice lacking beta-2-microglobulin ($B2m$), which have low-surface MHC-I expression and no MHC-I-educated NK cells. $B2m^{-/-}$ NK cells had a lower frequency of DNAM-1-expressing NK cells (40–45%), and DNAM-1$^+$ NK cells had a 50% reduction in DNAM-1 expression (Fig. 1a–c and Supplementary Fig. 1) when compared to wild-type mice.

**DNAM-1 expression correlates with educating impact of MHC-I.** Previously, we demonstrated that each individual MHC-I allele has a distinct educating impact on NK cells, with the D$^d$ allele having the strongest influence, followed by K$^b$ and a remarkably weak effect of D$^b$ (ref. 31). Here, DNAM-1 expression was assessed in mice with no MHC-I ($K^{b-/-}D^{b-/-}$) gene, a single-MHC-I allele ($K^{b-/-}$ or $D^{b-/-}$) and B6 wild-type ($K^bD^b$) mice. We found that about 45% of NK cells from $K^{b-/-}$ $D^{b-/-}{}^{-/-}$ mice were DNAM-1$^+$. Mice expressing only the weak educating MHC-I molecule D$^b$ did not have an increased DNAM-1$^+$ population. However, if one of the strong educating MHC-I molecules K$^b$ or D$^d$ was expressed, NK cells had a higher frequency of DNAM-1$^+$ cells (Fig. 1d). The increase in DNAM-1$^+$ cells thus followed the education hierarchy of the MHC-I alleles. In assessing DNAM-1 mean fluorescence intensity (MFI) on NK cells, expression of K$^b$ or D$^d$ alone, increased average DNAM-1 levels on the NK-cell pool (Fig. 1e). In conclusion, we found that the levels of DNAM-1 expression and the frequency of DNAM-1$^+$ NK cells correlated with the educating impact of the MHC-I allele.

**DNAM-1 expression on educated and uneducated NK cells.** We next hypothesized that MHC-I-educated and non-MHC-I-educated NK-cell subsets within the same mouse strain would also have differential expression of DNAM-1. We examined the DNAM-1 expression on NK-cell subsets as defined by their pattern of self-MHC-I-specific inhibitory receptor expression: Ly49A, Ly49C, Ly49G2, Ly49I and NKG2A (Supplementary Fig. 2a). This allowed us to concentrate on single-positive (SP) cells, and to assess the impact of one specific Ly49r on DNAM-1 expression in the absence versus the presence of NKG2A (Fig. 2a). We found that among NK cells expressing only one inhibitory Ly49r, the Ly49C-SP and Ly49I-SP had a higher proportion of DNAM-1$^+$ cells than the Ly49A-SP or Ly49G2-SP. The observed effect of Ly49r expression on DNAM-1 seemed to be additive, as the Ly49CI double-positive subset had a higher frequency of DNAM-1$^+$ cells than either of the Ly49C-SP or Ly49I-SP. This could also be seen among the corresponding subsets co-expressing NKG2A. Interestingly, all NKG2A$^+$ subsets had a higher frequency of DNAM-1$^+$ cells than NKG2A$^-$ subsets, with NKG2A-SP cells being 90% DNAM-1$^+$. However, with the addition of Ly49r, there was a small but consistent reduction in the expression of DNAM-1 on the NK-cell subsets. Thus, the more inhibitory Ly49r expressed by an NK cell in addition to NKG2A, the proportion of DNAM-1$^+$ cell was lower when compared to NKG2A-SP

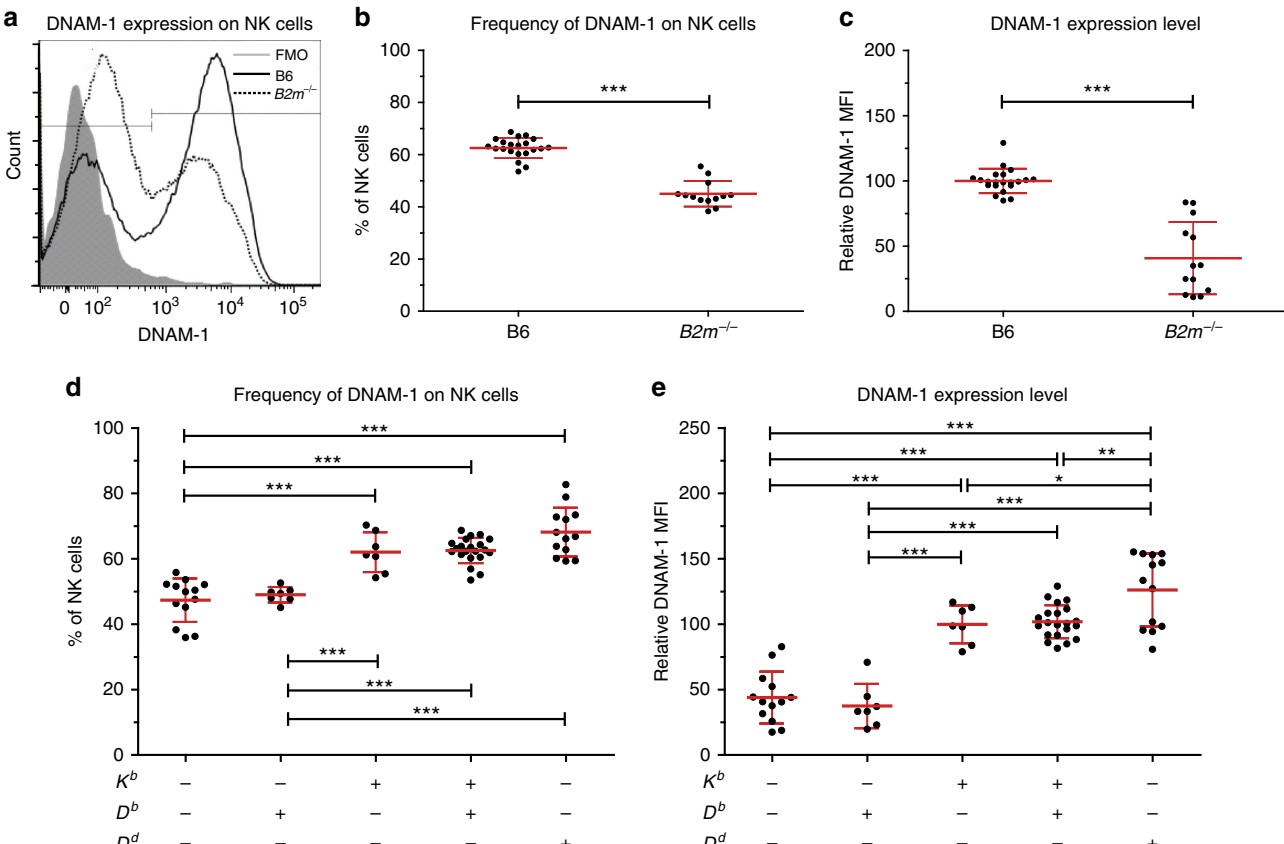

**Figure 1 | DNAM-1 expression is dependent on MHC-I expression.** (**a**–**c**) DNAM-1 expression on B6 wild type versus $B2m^{-/-}$ mice. (**a**) Representative histograms of DNAM-1 expression on B6 (black line), $B2m^{-/-}$ (dotted line) or FMO control (shaded grey). (**b**) Compilation of frequency of DNAM-1[+] NK cells and (**c**) compilation of DNAM-1 MFI on NK cells displayed as per cent of expression compared to B6 wild-type control. **b,c** represent five experiments with at least 2 mice per experiment and at least 13 mice in total. (**d,e**) DNAM-1 expression on mice with different MHC-I alleles. (**d**) Compilation of frequency of DNAM-1[+] NK cells and (**e**) compilation of DNAM-1 MFI on NK cells displayed as per cent expression compared to B6 wild-type control, both of three experiments with at least two mice per group and experiment. Significant differences are calculated by unpaired $t$-test (**b,c**) or one-way ANOVA (**d,e**), and are depicted as $*P < 0.05$, $**P < 0.01$, $***P < 0.001$. Error bars denote s.d.

NK cells. This was more prominent if inhibitory Ly49r that do not bind strongly to self-MHC-I were co-expressed with NKG2A (Ly49A and/or Ly49G2) and less pronounced when self-MHC-I-specific Ly49r (Ly49C and/or Ly49I) were co-expressed (Fig. 2a).

Interestingly, when we examined TIGIT, an inhibitory receptor for the DNAM-1-ligand CD155, we found that it is preferentially expressed on NKG2A[−] NK cells (Supplementary Fig. 3a), $K^{b-/-}D^{b-/-}$ NK cells had higher levels of TIGIT (Supplementary Fig. 3b) and the expression pattern on NK-cell subsets was opposite to that of DNAM-1 (Supplementary Fig. 3c), indicating that DNAM-1 and TIGIT may be regulated in a balanced fashion.

The higher expression of DNAM-1 on NK cells expressing self-MHC-I-binding inhibitory receptors could also be seen among other Ly49r-MHC-I allele combinations (Fig. 2b). Frequency and MFI of DNAM-1[+] cells among Ly49C-SP or Ly49I-SP was higher compared to Ly49A-SP or Ly49G2-SP NK-cell subsets in $D^{b-/-}$ mice (Fig. 2, Supplementary Fig. 2b). In D[d] single mice, there was also a significant increase of DNAM-1 expression, but in a different pattern based on self-educating NK-cell subsets in this host (Fig. 2b). In addition, the NK-cell subsets expressing Ly49A-SP or Ly49G-SP had the highest frequency of DNAM-1[+] NK cells among the NKG2A-negative subsets (Supplementary Fig. 2b). In $K^{b-/-}$ mice, none of the SP subsets exhibited a significant increase of DNAM-1 (Fig. 2b), although there was a tendency in the Ly49A-

SP NK cells, which could indicate a weak education impact of the D[b] molecule on this subset due to weak binding to MHC-I (ref. 32). In D[d]-transgenic mice ($K^bD^bD^d$), as expected, the Ly49A, Ly49C and Ly49G2-SP subsets had a higher expression of DNAM-1.

**Additive effect of multiple self-Ly49r on DNAM-1 expression.** Next, we asked whether NK cells with more inhibitory receptors express more DNAM-1, and whether any pattern emerging would depend on MHC-I expression. We divided NK-cell subsets of B6 and $B2m^{-/-}$ mice based on the number of inhibitory Ly49r and expression of NKG2A, which allowed us to identify and dissect two distinct patterns. DNAM-1 was associated with the number of Ly49r (regardless of specificity), both in the NKG2A[+] and NKG2A[−] subsets, albeit with a different impact. One component of this association is education-dependent (seen in NKG2A[−] subsets in B6), which can be described as 'the more Ly49r, the more DNAM-1' (Fig. 3a), irrespective of which Ly49r are expressed, as long as there is MHC-I expression. The more Ly49r that are expressed, the more likely it is, that this particular cell belongs to the educated pool of NK cells, thus corroborating our observation that DNAM-1 expression is associated with education.

A second component of the association between the number of Ly49r and DNAM-1 expression was education-independent (seen in $B2m^{-/-}$), which can be described as 'the more Ly49r, the less

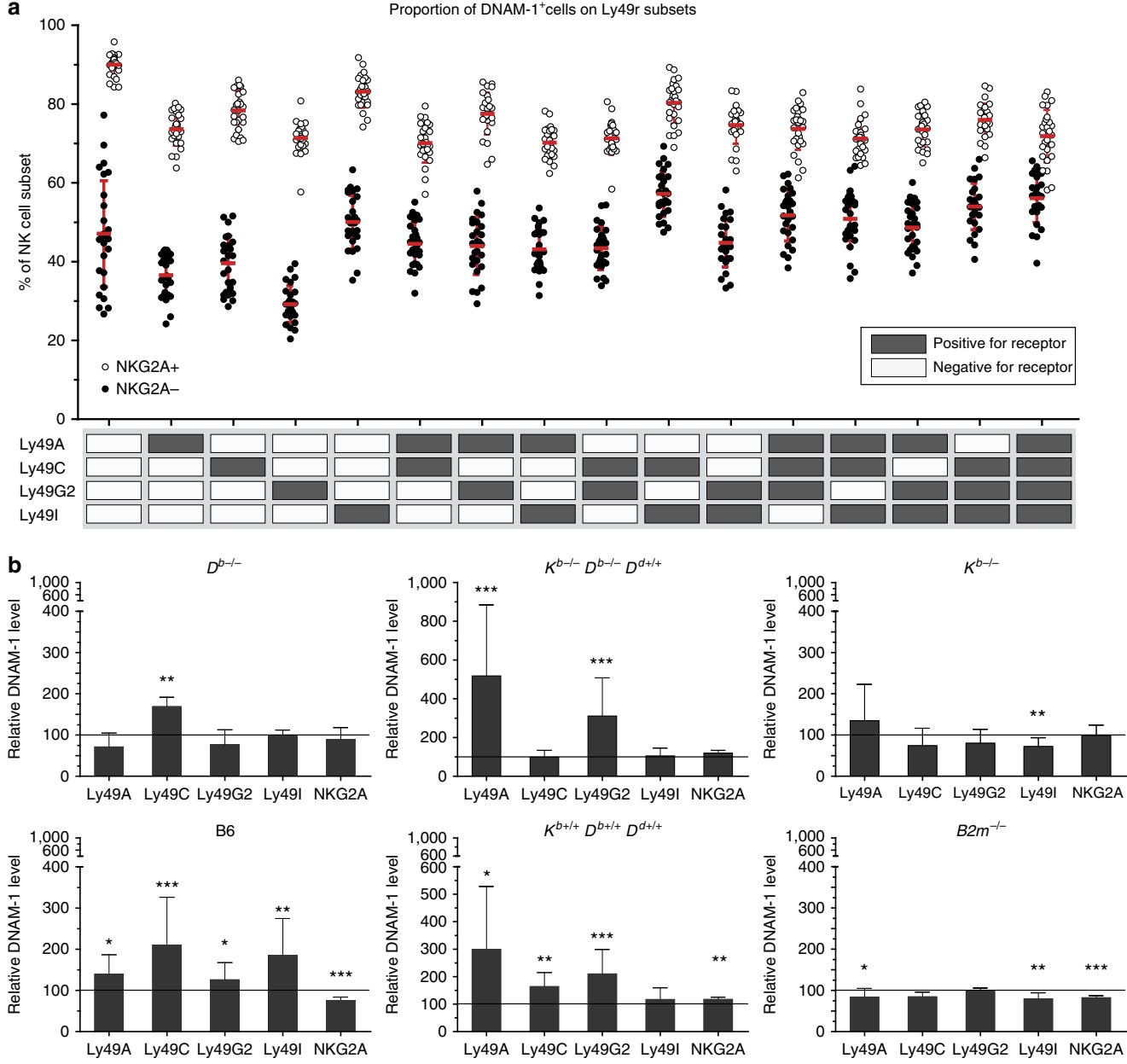

**Figure 2 | DNAM-1 expression correlates with inhibitory receptors for self-MHC-I.** (**a,b**) Frequency of DNAM-1$^+$ NK cells in B6 mice, (**a**) on NK-cell subsets that express a certain combination of inhibitory Ly49r in the presence (white dots) or absence (black dots) of NKG2A. Results are pooled from 26 mice taken from seven experiments. (**b**) DNAM-1 MFI on single-Ly49r$^+$ NK cells in $D^{b-/-}$, $D^d$-transgenic, $K^{b-/-}$, B6 ($K^b D^b$), $D^d$-transgenic ($K^b D^b D^d$) and $B2m^{-/-}$ mice. Results are shown as per cent of expression compared to the same NK-cells subset in $K^{b-/-} D^{b-/-}$ mice, which is represented by the line at 100% relative DNAM-1 expression. Data are from three experiments with at least six mice in total. Significant differences are calculated against the corresponding NK-cell subset in $K^{b-/-} D^{b-/-}$ mice by Mann–Whitney U-test and are depicted as *$P < 0.05$, **$P < 0.01$, ***$P < 0.001$. Error bars denote s.d.

DNAM-1' (Fig. 3b). This was most pronounced in the NKG2A$^+$ subset. We propose that this education-independent downregulation of DNAM-1 is associated with a higher probability of the cells being more mature, because NK cells acquire more Ly49r successively and cumulatively as they mature[33–35]. This is only seen in the presence of a strong educating MHC-I allele (Supplementary Fig. 4a–c) and is in line with the recent findings that DNAM-1 is downregulated upon maturation[29].

Recent models for NK-cell education are based on shifts in activation threshold determined by the combined input of inhibitory and activating receptors[4,36,37], where inhibitory

receptors decrease the activation threshold thereby increasing reactivity while activating receptors influence education in the opposite direction, that is, they tune down the responsiveness of an individual NK cell in continuous presence of the ligand for the activating receptor[37–39]. We therefore assessed the influence of an activating receptor (Ly49D) in the presence or absence of the cognate MHC-I allele (D$^d$) on DNAM-1 expression. We analysed NK cells that either express the inhibitory Ly49A or the activating Ly49D or both and found that the proportion of DNAM-1$^+$ cells was higher in the Ly49A-SP subset in D$^d$ single mice when compared to $K^{b-/-} D^{b-/-}$ or B6 mice. When Ly49D was expressed, the frequency of DNAM-1 was lower in the presence

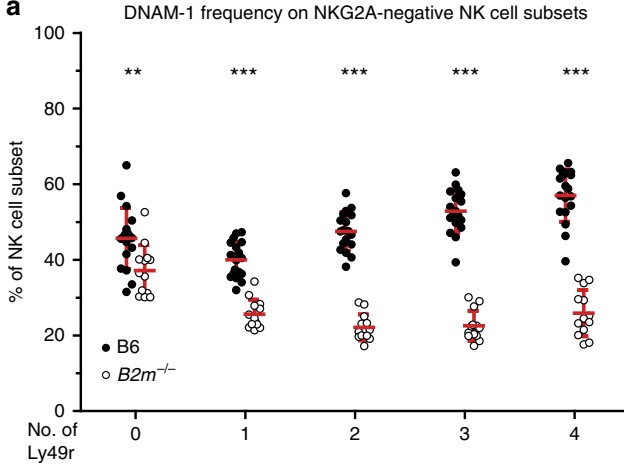

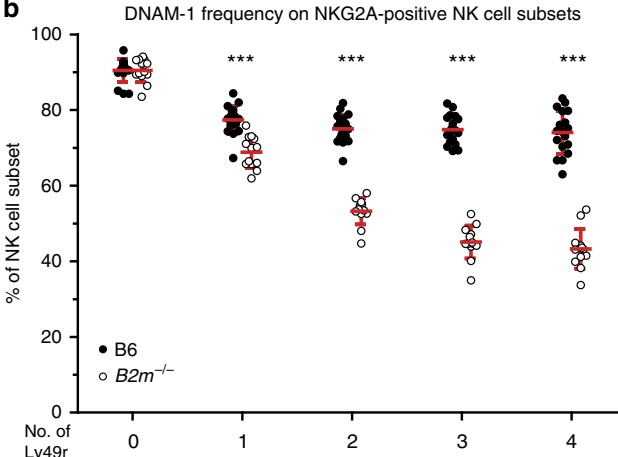

**Figure 3 | Correlation of DNAM-1 expression to the number of inhibitory Ly49r.** The number of inhibitory Ly49r on NK-cell subsets has an impact on the proportion of DNAM-1$^+$ cells within that subset. B6 or $B2m^{-/-}$ NK cells, stained for inhibitory receptors as for Fig. 2a, have been grouped based on the number of inhibitory Ly49r they express (regardless of which combination of inhibitory receptors). The frequency of DNAM-1$^+$ NK cells within each group is shown for NKG2A$^-$ cells in **a** and for NKG2A$^+$ cells in **b**. Significant differences are calculated comparing the corresponding NK-cell subset of B6 and $B2m^{-/-}$ mice by unpaired $t$-test and are depicted as **$P < 0.01$, ***$P < 0.001$. Error bars denote s.d.

of the ligand D$^d$ (Supplementary Fig. 4d). As previously published, both Ly49D and Ly49H were less frequent on DNAM-1$^+$ cells[28], while other activating receptors such as NK1.1, NKG2D, NKp46 and 2B4 had a higher expression of DNAM-1$^+$ NK cells compared to DNAM-1$^-$ NK cells within B6 mice (Supplementary Fig. 4e), or on total NK cells from B6 mice compared to $CD226^{-/-}$ mice (Supplementary Fig. 4f).

**Educating molecules correlate with DNAM-1 expression.** To dissect our data set in an unbiased way and to reduce the dimensionality, we performed a principal component analysis (PCA, Fig. 4) and an orthogonal projection and latent structures discriminant analysis (Supplementary Fig. 5). The PCA was calculated on the proportion of DNAM-1$^+$ cells within a given NK-cell subset based on the expression of inhibitory Ly49r and NKG2A in mouse strains which express one defined MHC-I allele. The PCA revealed that the frequency of DNAM-1$^+$ cells within each NK-cell subset separates these strains into three clusters (Fig. 4a) and identified three informative principal components

(PC, two first shown) that together account for 87% of the variance. PC 1 ($x$ axis) separates D$^d$ single (red) from $D^{b-/-}$ (blue) and $K^{b-/-}D^{b-/--/-}$ (yellow) and $K^{b-/-}$ mice (green). The difference between the three clusters in the PC 1-direction seems to be the presence of an MHC-I allele with a strong educating impact. $K^{b-/-}D^{b-/--/-}$ (yellow) and $K^{b-/-}$ mice (green) do not have an MHC-I allele that can measurably educate NK cells[31]. NK cells of $D^{b-/-}$ mice (blue) are educated by Ly49C/I, while NK cells in D$^d$ single mice (red) are educated by the presence of D$^d$ via Ly49A/G. Interactions between Ly49A and D$^d$ are of higher affinity, and confer a superior responsiveness as any other Ly49r-MHC-I pair[14,40,41]. For separation into PC 1 ($x$ axis), all subsets contribute positively (Supplementary Fig. 5a), while PC 2 ($y$ axis) separates D$^d$ single and $K^{b-/-}$ mice (lower quadrants) from $K^{b-/-}D^{b-/--/-}$ mice (cluster at 0) and $D^{b-/-}$ mice (upper quadrants). The main separator for PC 2 seems to be the possibility to receive education, either through Ly49C/I, or via Ly49A/G. This can be further supported by plotting the variables that contribute the separation in PC 2 (Fig. 4b), where NK-cell subsets expressing Ly49C/I exhibit positive values, while those with Ly49A/G have negative values. We conclude that the frequency of DNAM-1$^+$ cells among NK-cell subsets based on Ly49r expression can cluster the mouse strains according to known interactions of certain MHC-I alleles with specific Ly49r.

**DNAM-1$^+$ NK cells kill MHC-I-deficient tumour cells.** To test the hypothesis that DNAM-1-expressing NK cells represent the educated subsets endowed with capacity for missing self-recognition, we examined the ability of sorted DNAM-1$^+$NKG2A$^+$, DNAM-1$^-$NKG2A$^+$ and DNAM-1$^-$NKG2A$^+$ subsets of$^-$ IL-2-stimulated NK cells to kill MHC-I-expressing (RMA) and MHC-I-deficient (RMA-S) cells. DNAM-1$^+$NKG2A$^+$ and DNAM-1$^+$NKG2A$^-$ NK cells could differentiate between RMA and RMA-S cells (Fig. 5a–c), while DNAM-1$^-$NKG2A$^-$ NK cells could not discriminate between the two cell lines. RMA and RMA-S cells have similar but low levels of CD155 and CD112 (Supplementary Fig. 6), indicating that the difference in detection of missing self is due to a distinct difference in education and not a result of dissimilar stimulation via DNAM-1. All three populations could kill YAC-1 equally well (Fig. 5d), and are therefore functional in recognizing and killing tumour cells in a system that is primarily dependent on recognition by NKG2D. IL-2-stimulated NK cells from $CD226^{-/-}$ mice could discriminate between MHC-I$^+$ and MHC-I$^-$ target cells (Fig. 5e,f), which shows that the observed difference between DNAM-1$^+$ and DNAM-1$^-$ NK cells is not a result of direct interaction of DNAM-1 with the ligands on the target cells. This suggested that the presence of DNAM-1 on NK cells correlates with their ability for missing self-recognition but is not necessary to maintain that function. Along the same line, we observed that both DNAM-1$^+$ and DNAM-1$^-$ cells within educated subsets (Ly49A-SP in D$^d$ single mice, Ly49C-SP in $D^{b-/-}$ mice) performed equally well in the production of IFN-$\gamma$ and CD107 release in response to crosslinking of NK1.1 (Fig. 6). Thus, we conclude that DNAM-1 is not required to induce education.

**Lack of DNAM-1 does not impair missing self recognition.** On the basis of the rheostat model of education, NK cells are tuned by continuous interactions via their receptors with MHC-I and possibly other ligands. If DNAM-1 is crucial for reaching or maintaining the educated state, this could be tested by examining missing self-responses in mice lacking DNAM-1 or by blocking the interaction between DNAM-1 and it's ligands *in vivo*. $CD226^{-/-}$ mice rejected MHC-I-deficient target spleen cells, although slightly less efficiently compared to B6 wild-type mice

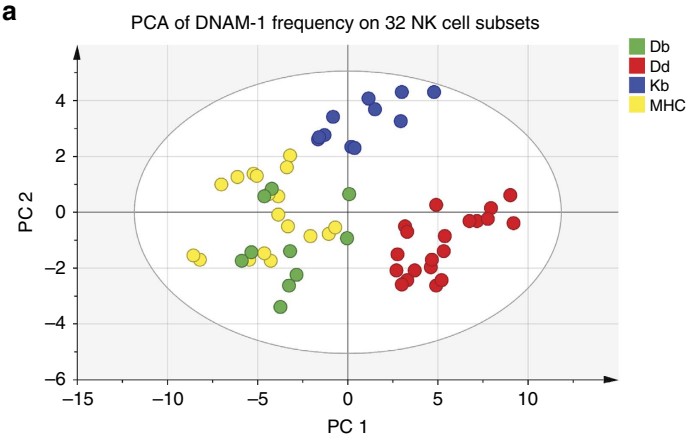

**a**

PCA of DNAM-1 frequency on 32 NK cell subsets

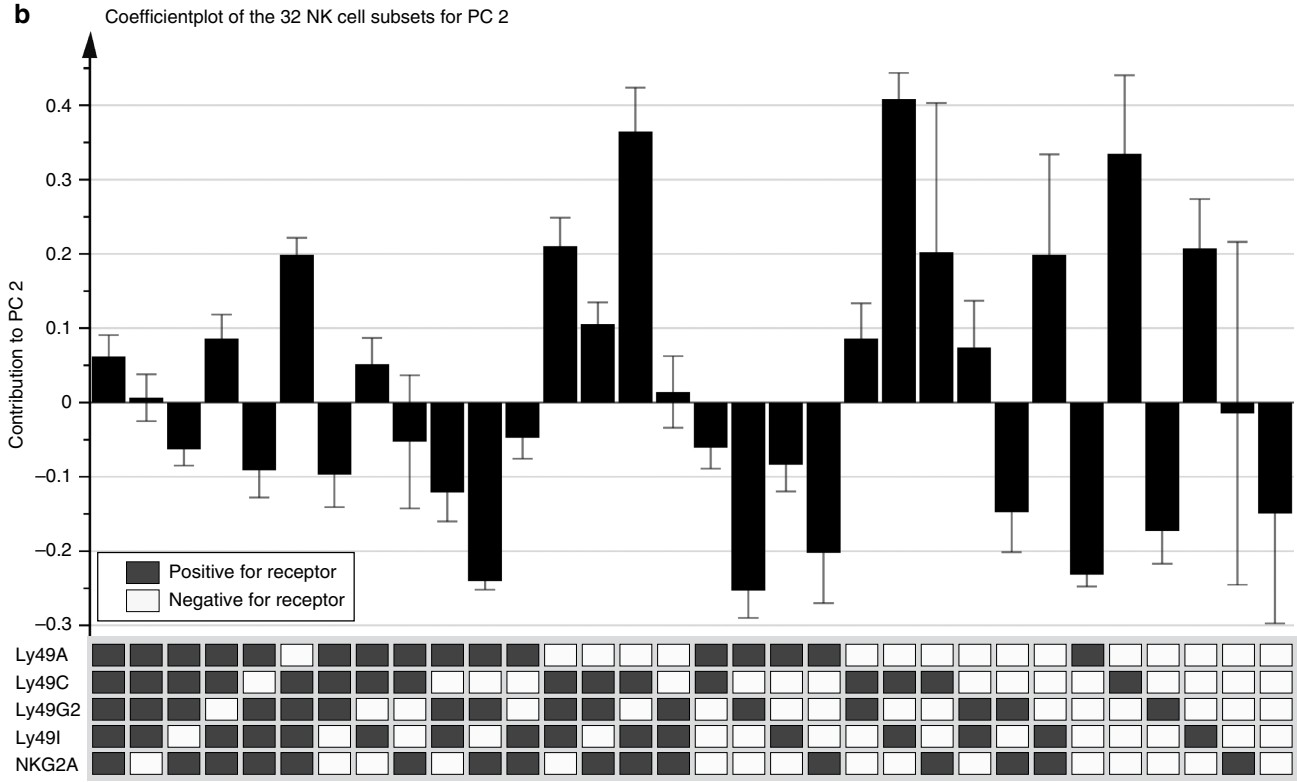

**b**

Coefficientplot of the 32 NK cell subsets for PC 2

**Figure 4 | PCA reveals that the mice spontaneously cluster into three main groups.** (**a**) PCA on $K^{b-/-}D^{b-/-}$, $D^{b-/-}$, $K^{b-/-}$ or $Kb^{-/-}$ $Db^{-/-}Dd^{+/+}$ based on the proportion of DNAM-1$^+$ cells (DNAM-1 frequency) on the 32 NK-cell subsets defined by inhibitory Ly49r and NKG2A expression. NK cells are gated on live singlets NK1.1$^+$CD3$^-$ cells. Data are pooled from eight independent experiments with at least eight mice per group. The model has three significant principal components (two first shown). x axis corresponds to PC 1, y axis to PC 2. (**b**) Variables contributing to the separation in predictive PC 2 for the PCA. Error bars represent the 95% confidence interval.

(Fig. 7a). This could indicate an education defect in the absence of DNAM-1, or that DNAM-1 is required for target cell killing. $CD226^{-/-}$ mice had a normal NK-cell compartment both in terms of frequency and absolute number[20]. Nevertheless, a reduction of NKG2A-SP cells could be observed in $CD226^{-/-}$ mice when compared to B6 wild-type mice (Fig. 7b). To examine whether blockade of DNAM-1 can change the education status of NK cells, B6 mice were treated with anti-DNAM-1 monoclonal antibody for 2 or 14 days[42]. Treatment completely abrogated binding of several DNAM-1 antibodies (clones 10E5, 480.1 and TX-42.1; Supplementary Fig. 7) thus verifying that we could maintain complete blockade of DNAM-1 in vivo. Treated mice had no change in frequency and number of total NK cells and in vivo blocked NK cells could still efficiently

kill MHC-I-deficient spleen target cells (Fig. 7c). We observed only small changes in the Ly49r repertoire, however, similar to $CD226^{-/-}$ mice, the frequency of NKG2A-SP NK cells was markedly reduced at both time points tested (Fig. 7d). This demonstrates that DNAM-1 is neither necessary to achieve education nor to maintain the educated state of mature NK cells, and that DNAM-1 is not the major determinant for the rejection of MHC-I$^-$ spleen cells.

**Expression of DNAM-1 during NK-cell development.** The notion that DNAM-1 expression may be related to education raises the question as to which stage during development of NK

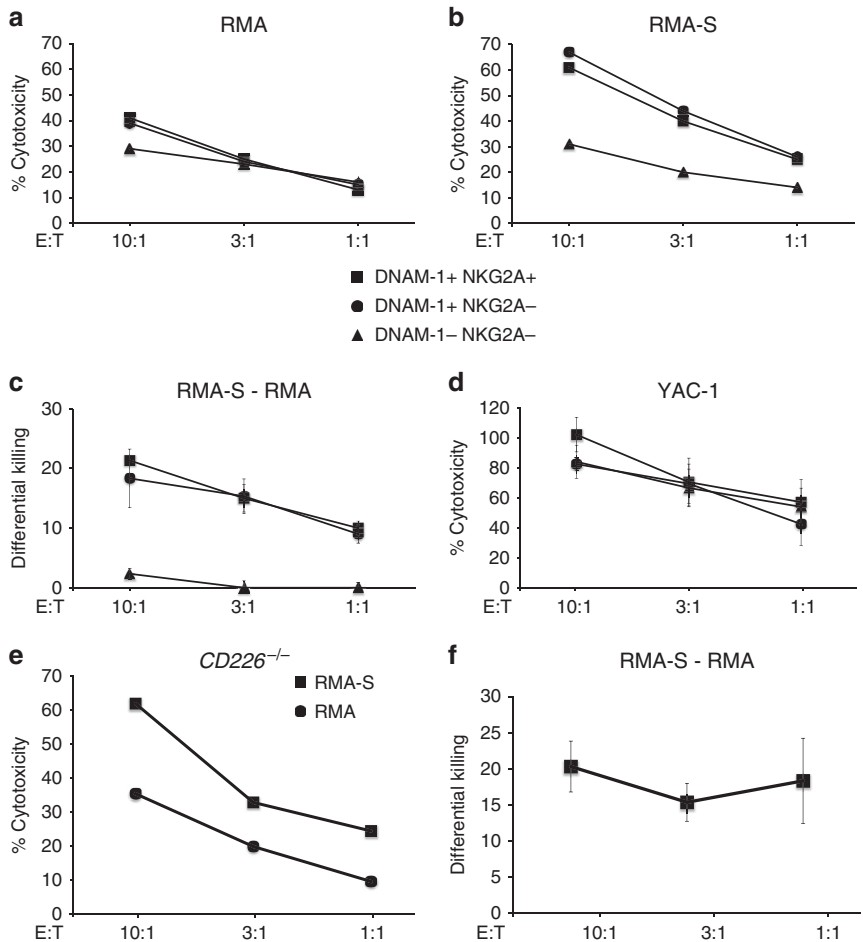

**Figure 5 | DNAM-1 is not required for missing self recognition.** The ability of NK cells to kill MHC-I-deficient tumour targets correlates with DNAM-1 expression. (**a–d**) *In vitro* cytotoxicity of sorted DNAM-1$^+$NKG2A$^+$ (squares), DNAM-1$^+$NKG2A$^-$ (circles) and DNAM-1$^-$NKG2A$^-$ (triangles) IL-2-stimulated NK cells. (**a**) Killing of RMA, one representative of three experiments shown. (**b**) Killing of RMA-S, one representative of three experiments shown. (**c**) Calculation of the differential killing ( ± s.d.) between RMA-S and RMA killing, $n = 3$ experiments. (**d**) Killing of YAC-1 $n = 3$ experiments. (**e,f**) *In vitro* cytotoxicity towards RMA (circles) and RMA-S (squares) by IL-2-stimulated $CD226^{-/-}$ NK cells. (**e**) Shows one representative experiment of three and (**f**) shows differential killing between RMA-S and RMA, $n = 3$ experiments. Error bars denote s.d.

cells DNAM-1 expression first occurs. During differentiation, common lymphocyte progenitors (CLP, lin$^-$c-kit$^-$flk2$^+$CD27$^+$CD224$^+$IL-7Rα$^+$CD122$^-$, Supplementary Fig. 8) develop into pre-NK precursors (pre-NKP, lin$^-$c-kit$^-$flk2$^-$CD27$^+$CD224$^+$IL-7Rα$^+$CD122$^-$), progress into refined NKP (rNKP, lin$^-$c-kit$^-$flk2$^-$CD27$^+$CD224$^+$IL-7Rα$^+$CD122$^+$) and ultimately give rise to NK1.1$^+$NKp46$^+$ NK cells, which egress from the bone marrow (BM)[43]. Both DNAM-1 and NKG2A were absent on CLP, a few pre-NKP started to express DNAM-1 while rNKP were mostly positive for DNAM-1 (Fig. 8a,b). Expression of NKG2A, the earliest MHC-I-specific inhibitory receptor[44], was detected first on NK1.1$^+$ immature BM NK cells.

Therefore it appears that DNAM-1 expression is switched on before and independently of MHC-I-specific inhibitory receptors, opening the possibility that it may be involved in educating NK cells already in the BM. DNAM-1 expression was similar on NK cells from spleen, BM and liver from 10-day-old B6 and $K^{b-/-}D^{b-/-}$ mice, before the onset of Ly49r expression (Supplementary Fig. 9), suggesting that the reduced level seen in adult $K^{b-/-}D^{b-/-}$ mice is imposed later during life. CD27 and CD11b are markers that dissect NK cells into functionally distinct maturation subsets[45]. DNAM-1 expression increases when NK cells upregulate CD27, and DNAM-1 expression is highest on CD27$^+$CD11b$^-$ NK cells and then goes down again upon further maturation to CD27$^-$CD11b$^+$ cells (Fig. 8c,d).

**MHC-I-dependent plasticity of DNAM-1 expression.** Adoptive transfer of NK cells from mice with MHC-I expression to recipients that lack MHC-I can lead to re-education of the NK cells and alteration in expression of activating receptors and adhesion receptors[36,46,47]. Since we observed that $B2m^{-/-}$ mice have reduced levels of DNAM-1, we hypothesized that the expression of DNAM-1 would change on NK cells upon adoptive transfer of mature NK cells to a different MHC-I environment. Indeed, there was a marked reduction in DNAM-1 expression on B6 NK cells transferred to $B2m^{-/-}$ mice (Fig. 9b). This effect was even more pronounced when we stained for inhibitory receptors and gated on the Ly49C-SP or Ly49I-SP while no reduction could be seen in the other SP subsets, including the NKG2A-SP (Supplementary Fig. 9). Transfer of mature $B2m^{-/-}$ NK cells to RAGcγ$^{-/-}$ mice (as MHC-I$^+$ hosts) resulted in upregulation of DNAM-1 MFI on the transferred NK cells that

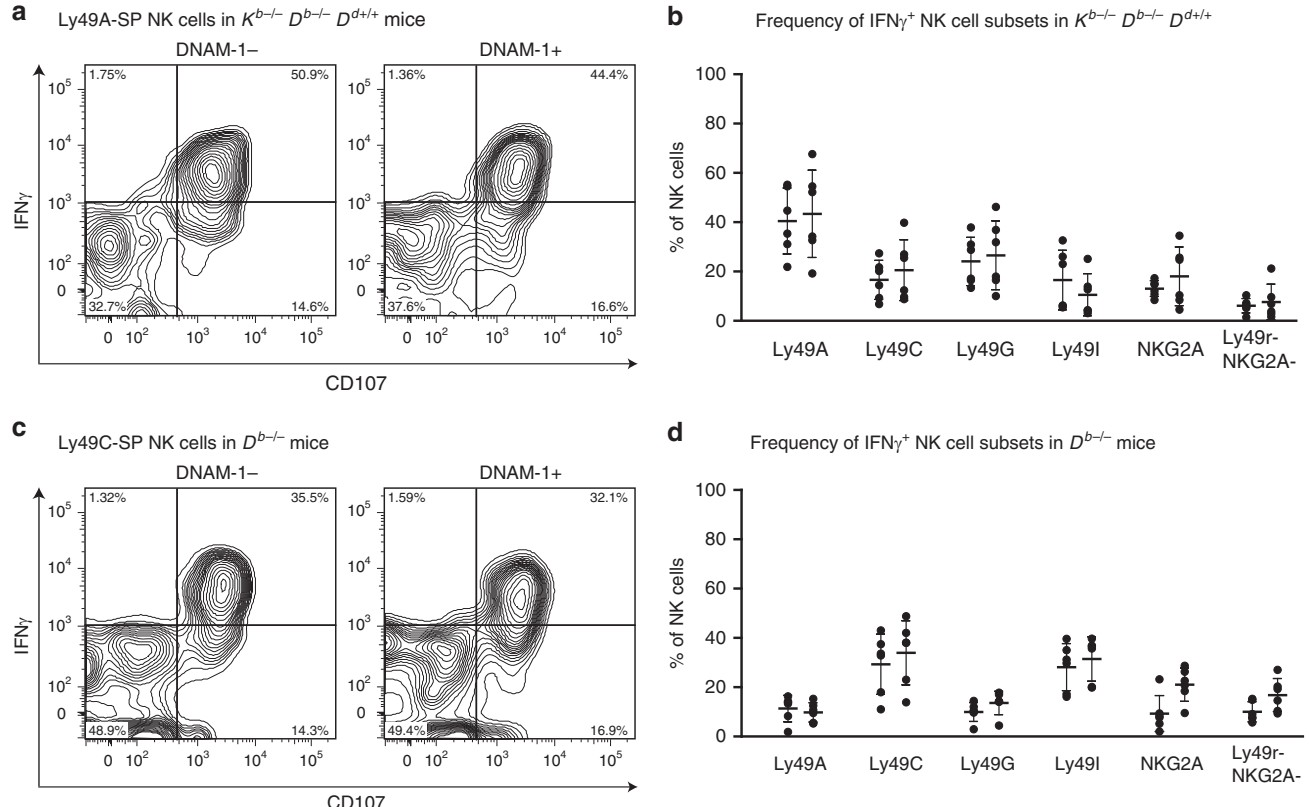

**Figure 6 | DNAM-1$^+$ and DNAM-1$^-$ NK cells respond to receptor crosslinking.** (**a**) Expression of IFN-γ and CD107a on DNAM-1$^+$ and DNAM-1$^-$ Ly49A SP NK cells from a $D^d$-transgenic mouse on $K^{b-/-}D^{b-/-}$ background ($K^{b-/-}D^{b-/-}D^{d+/+}$), after crosslinking with anti-NK1.1, one representative plot shown. (**b**) Frequency of IFN-γ$^+$ NK cells on Ly49r SP subsets ($n = 6$, 3 experiments). (**c**) Expression of IFN-γ and CD107a on DNAM-1$^+$ and DNAM-1$^-$ Ly49C-SP NK cells from a $D^{b-/-}$ mouse ($K^{b+/+}D^{b-/-}$) after crosslinking with anti-NK1.1, one representative plot shown. (**d**) Frequency of IFN-γ$^+$ NK cells on Ly49r SP subsets ($n = 6$, 3 experiments).

expressed DNAM-1 (Fig. 9b). Transfer of $B2m^{-/-}$ NK cells to $B2m^{-/-}$ mice, however, resulted in a slight reduction of the DNAM-1$^+$ population without a change of MFI on the cells that were DNAM-1$^+$. No significant changes in the frequency of DNAM-1$^+$ NK cells were observed in either transfer situation (Fig. 9c). Thus, while DNAM-1 expression is an early event in the development of NK cells, its expression is plastic and can be influenced by the MHC-I environment.

## Discussion

In the present study, we have investigated a possible link between DNAM-1 expression and education state of NK cells and observed two interesting patterns. The majority of NK cells expressing NKG2A had high levels of DNAM-1. This was observed even in mice deficient for MHC-I molecules, indicating that this is not the result of signals associated with MHC-I-dependent education, but rather due to common regulation of DNAM-1 and NKG2A at the level of the individual cell. Furthermore, analysing the NKG2A$^-$ cells, we could detect a close association between DNAM-1 expression and Ly49r-expression pattern in the remaining NK populations. This association was strongly influenced by the presence of host MHC-I ligands for each Ly49r, and it was manifested both in terms of a higher frequency of NK cells expressing DNAM-1, and higher average expression levels in the DNAM-1$^+$ NK cells. Given the MHC-I-dependency of this pattern, and our observation that DNAM-1 expression changed upon transfer of mature NK cells from $B2m^{-/-}$ mice to an environment expressing the

critical MHC-I molecule (or vice versa), we conclude that regulation of DNAM-1 expression is controlled by signals initiated via Ly49r-MHC-I interactions in the education process. Several critical signalling events induced by inhibitory receptors have been suggested to be important for education, including SHP-1-dependent dephosphorylation of LAT, PLCγ1/2 and VAV-1 (refs 48,49), c-Abl-mediated dephosphorylation of the adaptor protein Crk[50] and Cbl-mediated ubiquitination of LAT[49]. Furthermore, inositol phospholipid signalling in NK-cell biology seems to play a role in education[10]. Interestingly, Gumbleton et al.[10] observed that SHIP-1$^{-/-}$ NK cells, exhibit a reduced proportion of DNAM-1$^+$ NK cells, and a reduced capacity to reject allogeneic BM cells and a receptor repertoire reminiscent of that of $K^{b-/-}D^{b-/-}$ mice. DNAM-1$^+$ (but not DNAM-1$^-$) NK cells from normal B6 mice could sense the absence of MHC-I on tumour target cells and perform missing self-killing in vitro. All these observations suggest a strong association between DNAM-1 expression and education state, even when the latter is altered in mature NK cells in a manner consistent with the 'rheostat model'.

There are at least three possible interpretations offering simplistic models. First, DNAM-1 expression on NK cells may be a pre-requisite for education to occur in vivo, marking NK cells that are 'educate-able'. Second, DNAM-1 may be a major functional determinator of education, similar as proposed by Enqvist et al.[25], that is, induced by education and required to maintain that state. The third interpretation is that DNAM-1 does neither induce nor control education, but rather is a downstream correlate of education.

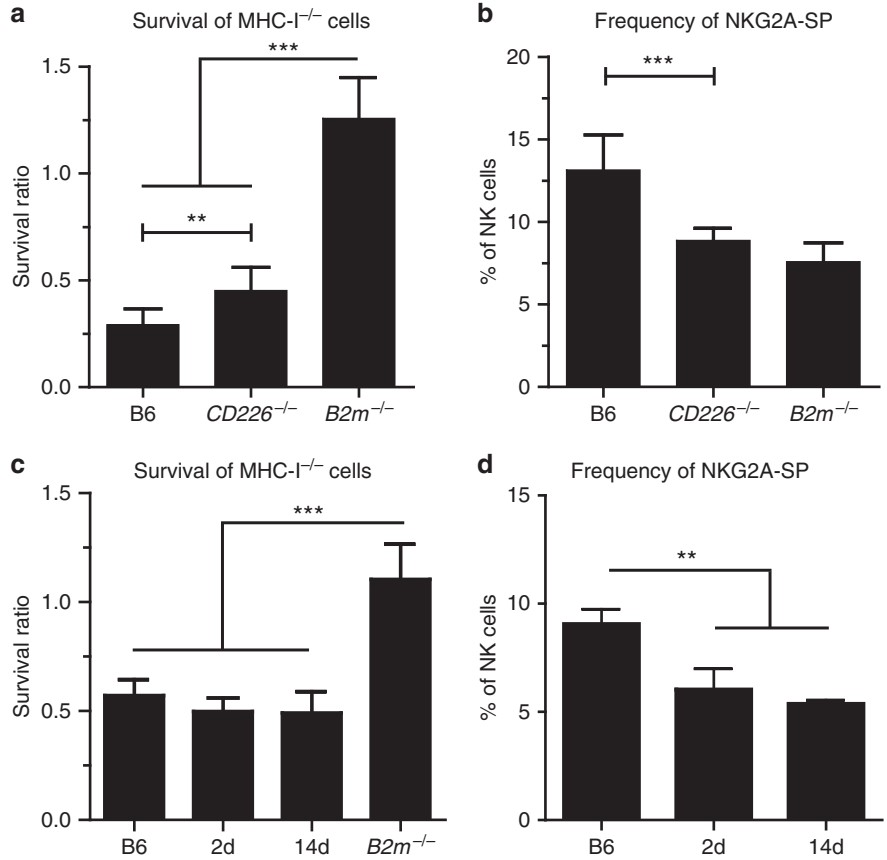

**Figure 7 | Absence of DNAM-1 does not abrogate missing self killing.** *In vivo* killing of MHC-I-deficient spleen cells were assessed in $CD226^{-/-}$ mice (**a**) or B6 wild-type mice after treatment with DNAM-1 mAb (**c**). CFSE-labelled B6 or MHC-I$^{-/-}$ spleen cell suspensions were inoculated i.v. and rejection was assessed 44 h later in the spleen. (**b**) Frequency of NKG2A-SP NK cells gated on live, singlet CD3$^-$NK1.1$^+$Ly49r$^-$ cells in $CD226^{-/-}$ mice. Bar graphs show data from six (B6, $B2m^{-/-}$), seven (2 days) or four (2 weeks) mice per group of two independent experiments with at least two mice per group. (**c,d**) DNAM-1 was blocked by injection of 200 μg of anti-DNAM-1 (clone 3B3) for 2 days or 2 weeks every 5 days and rejection of CFSE-labelled B6 or MHC-I$^{-/-}$ spleen cells was measured 44 h after inoculation of target cells (**d**) Frequency of NKG2A-SP NK cells gated on live, singlet CD3$^-$NK1.1$^+$Ly49r$^-$ cells in B6 mice treated with anti-DNAM mAb for 2 days or 2 weeks. Bar graphs show data from 6 (B6), 4 ($B2m^{-/-}$) or 14 ($CD226^{-/-}$) mice per group of two independent experiments with at least two mice per group. *P* values are calculated with *t*-test and are depicted **P<0.01, ***P<0.001. Error bars denote s.d.

These models are not mutually exclusive. However, our results are most consistent with the third possibility. It is first important to note that in comparison with the previous literature, our data adds several important pieces of information, apart from the demonstration of an association between DNAM-1 expression and MHC-I-dependent education also in the mouse. One is that DNAM-1 is expressed highly at an early stage in the BM, before several other important NK-cell receptors. This implies that DNAM-1 may be important for cellular development and the acquisition of functional properties during this process. A second novel conclusion from our study is that DNAM-1 is not required for either becoming MHC-I-educated or perform missing self-recognition per se, as most clearly demonstrated by our results with $CD226^{-/-}$ mice as well as mice treated with prolonged DNAM-1 blockade. In their study of human NK cells, Enqvist *et al.*[25] demonstrated that DNAM-1 and LFA-1$^{open}$ co-localize at the immune synapse and that the expression levels of both DNAM-1 and LFA-1$^{open}$ are higher in educated NK cells. However, that study did not assess whether DNAM-1 is required for killing or education. Our data indicate that at least in the mouse, DNAM-1 is associated with, but not required for MHC-I-dependent education of NK cells.

It is therefore reasonable to discuss a more complex hypothesis, considering also a recent model for regulation of DNAM-1 expression during NK-cell development[29], which proposes that murine NK cells gradually lose DNAM-1 during maturation, eventually resulting in a DNAM-1$^-$ population. This downregulation of DNAM-1 was shown to be at least partly regulated at the mRNA level[28,29], but regulation due to exposure to the ligand may also occur[51]. Our data on the early onset of DNAM-1 expression on NK-cell precursors in the BM fits with this model, and so does our data on generally lower expression at later maturation stages. However, our observations add new important features to the model, for example, that other patterns are superimposed on this gradual decline in DNAM-1 expression (Fig. 10). One is the maintained expression of DNAM-1 on NKG2A$^+$ cells (event 1 in Fig. 10). This is likely not directly related to the recently proposed concept of 'two schools of NK education'[52], one driven by NKG2A and the other by killer-cell immunoglobulin-like receptors, since this association between DNAM-1 and NKG2A occurred also in $B2m^{-/-}$ animals. However, it may influence the function of cells expressing NKG2A, regardless of whether they have been educated through this receptor. Moreover, our data on adoptively transferred mature NK cells indicate that re-tuning may be regulated

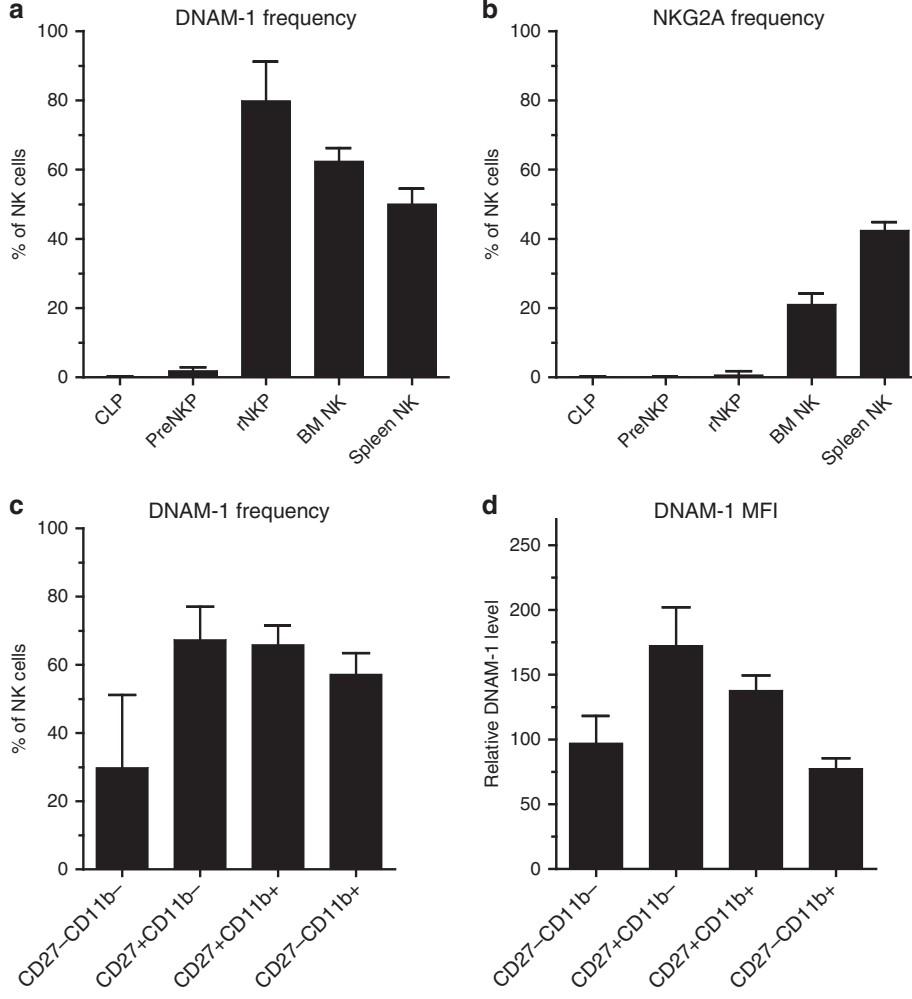

**Figure 8 | DNAM-1 is expressed in early NK-cell progenitors before inhibitory receptors.** (**a**) DNAM-1 expression, (**b**) NKG2A expression on CLPs, pre-NKPs, R-NKPs and mature NK cells. Gating strategy for CLP pre-NKP, and rNKP is shown in detail in Supplementary Fig. 8. Mature BM or splenic NK cells were defined by gating on NK1.1$^+$CD3$^-$ cells. NKG2A or DNAM-1 expression was analysed in these subsets. Pooled data from at least three different experiments from 14 mice are shown, with s.d. (**c**) DNAM-1 frequency and (**d**) DNAM-1 MFI on NK cells of different maturation stages in the spleen. Pooled data from at least three different experiments from 12 mice are shown with s.d. Error bars denote s.d.

differently depending on which receptor class delivers the inhibitory signals.

Second, education via interactions of inhibitory Ly49r with self-MHC-I seems to restrain the developmental downregulation of DNAM-1 (event 2 in Fig. 10), as NK cells that express a self-specific Ly49r (for example, Ly49C in B6 mice) express more DNAM-1 per cell and have a higher proportion of DNAM-1$^+$ cells. Education may even act to increase DNAM-1 expression, as seen in the adoptive transfer experiments, and also in the transition from the CD27$^-$CD11b$^-$ to CD27$^+$CD11b$^-$ stage. At this maturational stage NK cells experience a proliferation burst, begin to express Ly49r and start to sense MHC-I (ref. 45). Furthermore, our data using CD226$^{-/-}$ mice and the in vivo blocking experiments reveal that NK-cell education can occur and be maintained independently of DNAM-1.

In this more complex model, DNAM-1 would not be a major mechanistic determinator of education, but it may represent a molecule that endows educated NK cells with additional functional features that provide cells with increased functional capabilities. This does not exclude that other events may also contribute to maintain or enhance DNAM-1 expression. Interestingly, Nabekura *et al.*[27] observed that DNAM-1

expression increased during differentiation of activated NK cells in MCMV infection and that DNAM-1 may have a functional role for the generation of long-lived memory NK cells.

DNAM-1 expression early in NK-cell development may contribute, but not be essential, to promote cellular interactions required for education. One likely candidate as an educator of NK cells are dendritic cells (DCs) or BM stromal cells, as they establish crosstalk with NK cells that reciprocally regulate their functions[53,54]. DNAM-1 could be critical to support NK-cell–DC interactions or promote migration to sites where these interactions occur. Interactions of NK cells via DNAM-1 does not result in lysis of normal cells, likely because healthy cells express normal levels of self-MHC-I that inhibit killing upon interaction with inhibitory receptors on the NK cell. Thus, in the absence of effective inhibitory receptor signalling, the engagement of DNAM-1 and other triggering receptors may result in induction of cytotoxicity. Furthermore, NK cells may use DNAM-1 while they egress from the BM, and therefore the first maturation stage in the spleen may have lower levels due to ligand-induced downregulation. Interestingly, the CD27$^+$CD11b$^-$ stage, when DNAM-1 levels are elevated, coincides with the maturational stage in which NK cells

 

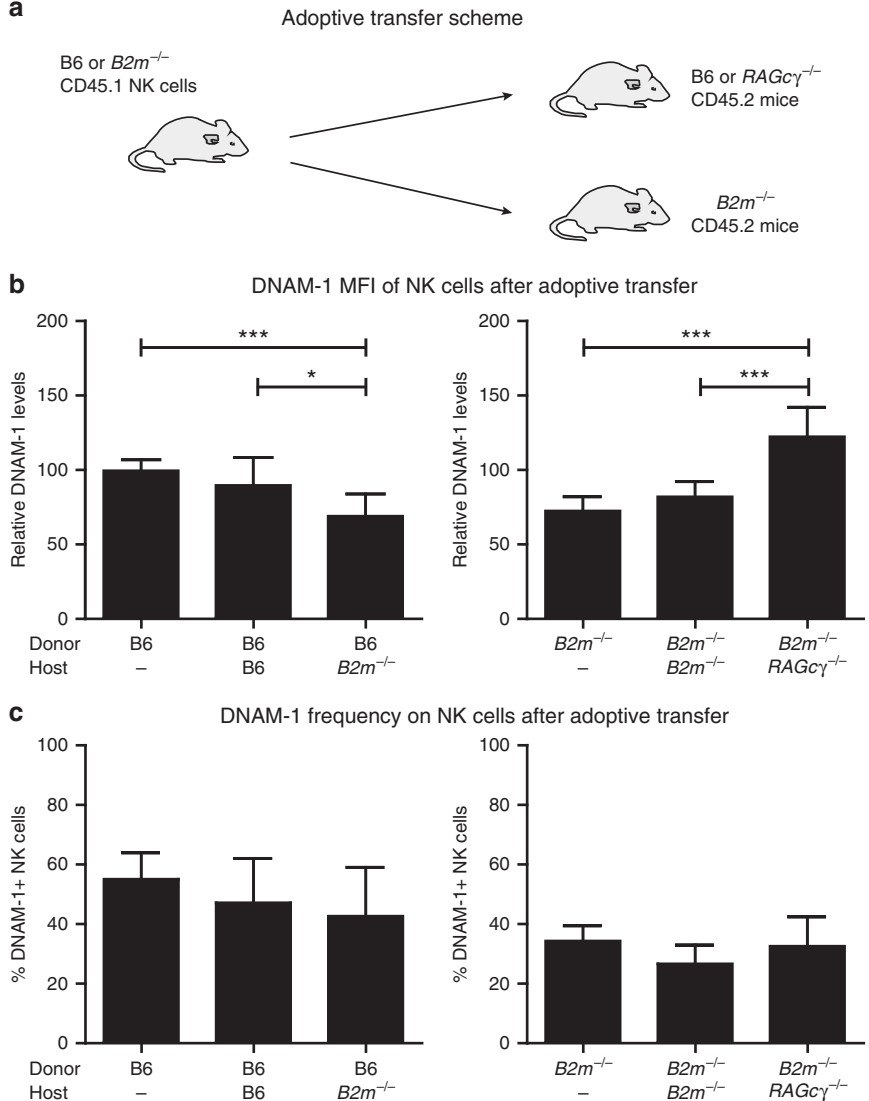

**Figure 9 | DNAM-1 expression changes after transfer to an MHC-I-different environment.** Expression of DNAM-1 changes after adoptive transfer from B6 to B6 ($n=8$) or $B2m^{-/-}$ ($n=9$) mice or from $B2m^{-/-}$ to $B2m^{-/-}$ ($n=7$) or MHC-I[+] RAGc$\gamma$ ($n=8$) mice. (**a**) Mature spleen NK cells were transferred as previously described[36] (**b**) DNAM-1 frequency and (**c**) DNAM-1 MFI on NK cells after adoptive transfer. NK cells from transferred mice are gated on CD45.1[+]NK1.1[+]CD3[−] cells. Data are pooled from three independent experiments. $P$ values are calculated using $t$-test for frequencies or Mann–Whitney test for MFI and are depicted *$P<0.05$, ***$P<0.001$. Error bars denote s.d.

experience a proliferation burst. Higher levels of DNAM-1 during this stage could enable proliferating NK cells to adhere to other cells and test their Ly49r against the host MHC-I molecules. Intriguingly, as NK cells progress from the CD27[high] via the double positive to the CD27[low] stage the Ly49r repertoire gradually shifts towards expression of self-specific Ly49r in the later maturational stages[45].

DNAM-1[+] NK cells could kill MHC-I-deficient tumour cells more efficiently than DNAM-1[−] NK cells. Whether this result shows that DNAM-1[−] NK cells are not sufficiently educated to perform target cell killing, or if they are defective in NK-cell–target cell interactions, is unclear. On the one hand, IL-2-stimulated $CD226^{-/-}$ NK cells could perform missing self-reactions against tumour cells *in vitro,* while on the other hand $CD226^{-/-}$ mice exhibited slightly reduced capacity for missing self rejection of MHC-I[−] cells *in vivo.* However, we cannot exclude that this difference is due to a mechanism distinct from target recognition, for example, homing or migration. We can conclude that while DNAM-1[+] NK cells can perform missing

self recognition superior to DNAM-1[−] NK cells, DNAM-1 itself is not essential for NK-cell education.

Does this model apply also to DNAM-1 expressed by human NK cells? This will require further studies. On the one hand, all NK cells express DNAM-1 in the human, unlike in the mouse[25,29]. However, this may represent a quantitative rather than qualitative difference between the species, as there is evidence for reduced levels of DNAM-1 with increased age and in disease in the human[55,56]. On the other hand, there is some evidence for a direct role for DNAM-1 in the immune synapse formed by human NK-cells, and our data on NK cells in KO mice does not exclude a more direct role for DNAM-1 in human NK-cell education or NK-cell–target interaction.

We conclude that high DNAM-1 expression is closely associated to NK-cell education, but not necessary to reach or maintain the educated state in the mouse, nor to perform missing self-recognition. Based on our own data as well as those published by others[25,27,29], we propose a model in which DNAM-1 expression is turned on early during NK-cell development in

 

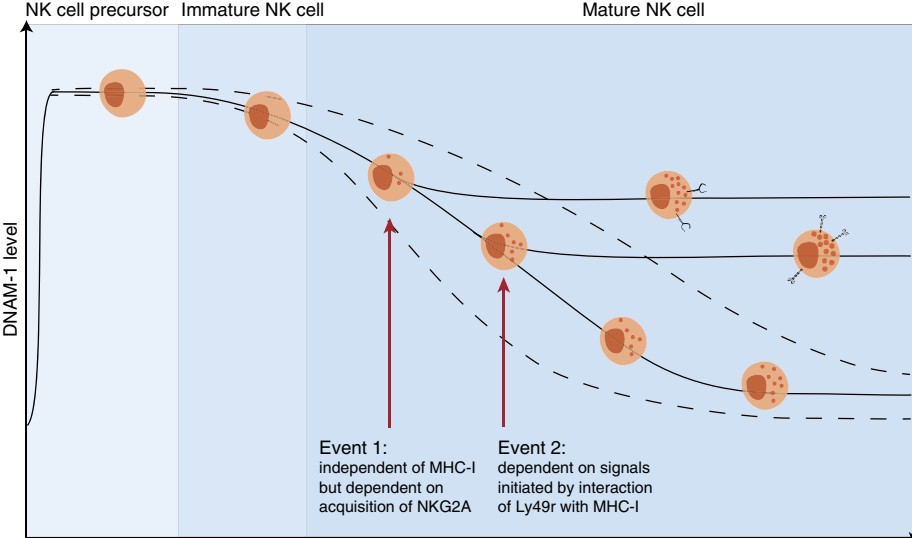

**Figure 10 | Model for expression of DNAM-1 during NK-cell development and education.** Expression of DNAM-1 is initiated by NK-cell precursors in the BM, before the onset of MHC-I-specific inhibitory NK-cell receptors. During development, NK cells then gradually decrease DNAM-1 levels[29], which ultimately may result in DNAM-1$^-$ NK cells. NK cells that start to express either NKG2A (event 1) or inhibitory Ly49r specific for self-MHC-I (event 2) are rescued from maturation-induced loss of DNAM-1. Event 1 can occur without event 2 and vice versa. Continuous interactions between inhibitory Ly49r and MHC-I retain DNAM-1 expression and prevent complete loss of DNAM-1. DNAM-1 may be important for NK-cell development and beneficial for educated NK cells during distinct maturation stages and in interactions with surrounding cells. This model does not exclude that later events associated with differentiation and proliferation of NK cells[27] can also maintain or even induce an increase in DNAM-1 levels/expression (not shown in the figure).

the BM, but is then gradually lost so that some NK cells eventually become DNAM-1$^-$ (Fig. 10). However different events such as expression of NKG2A, or self-MHC-I-recognizing Ly49r may prevent further loss of DNAM-1 in individual NK cells, and may even contribute to increased DNAM-1 expression. DNAM-1 in itself is not the major mechanistic determinator of the educated state, but we propose that increased levels of this molecule contribute to the functional performance of NK cells. Early during development, DNAM-1 may contribute to NK-cell migration to specific niches and there further assist in interactions with educating cells. Once in the educated state, NK cells may use DNAM-1 for migration into inflamed tissues and interact with putative target cells. It may also facilitate regular interactions with DCs, which are used by NK cells to maintain baseline effector functions[53]. Finally, DNAM-1 expression may be important in tuning both activity and homeostasis of NK cells upon activation and proliferation in response to different infections and of memory NK cells. Although there is still no molecular marker for the educated state, the emerging pattern is that NK cells educated on MHC-I have increased expression of at least two adhesion molecules, DNAM-1 and KLRG1 (refs 36,57), which contribute to optimize the function of educated NK cells.

## Methods

**Mice.** Mice were maintained at Karolinska Institutet, at the animal facility of the Department of Microbiology, Tumor and Cell Biology. Experiments were performed according to governmental and institutional regulations and were approved by the north Stockholm district court (Stockholms Norra djurförsökse-tiska nämnd). Animals were 6–12 weeks of age at the start of the experiments. Both male and female mice were used. C57BL/6 (B6) and CD45.1$^+$C57BL/6 (B6Ly5.1) mice were bred for at least 10 years at KI. Beta-2-microglobulin ($B2m^{-/-}$)[58], $K^{b-/-}D^{b-/-}$ (ref. 59), RAG2$^{-/-}$ common gamma$^{-/-}$ (RAGcγ)[60] and D$^d$-transgenic[61] were backcrossed to B6 for at least ten generations before intercross. $B2m^{-/-}$ mice were crossed to CD45.1$^+$C57BL/6 to generate CD45.1$^+$ $B2m^{-/-}$ mice used for transfer of non-MHC-I-educated NK cells. CD226$^{-/-}$ mice were described previously[20]. As MHC-I-deficient target cells in the *in vivo* cytotoxicity assay, spleens from $B2m^{-/-}$ or $K^{b-/-}D^{b-/-}$ mice were used and are referred to as MHC-I$^-$ cells.

**Antibodies and flow cytometric analysis.** For flow cytometry, single-cell suspensions were labelled with fluorescently conjugated anti-mouse antibodies. All antibodies were titrated and used at optimal dilutions. CD3 (145.2C11), CD27 (29A1.4), CD11b (M1/70.15), CD69 (H1.2F3), IFN-γ (XMG1.2), CD244 (m2B4(B6)458.1), cKit (2B8), CD127 (A7R34), CD122 (TM-β1), CD135 (A2F10), NK1.1 (PK136), KLRG1 (2F1), CD107a (1D4B), NKp46 (29A1.4), DNAM-1 (10E5, 480.1, TX-42.1 and 3B3 (ref. 28)) and Ly49A (YEI/48) (Biolegend), Ly49G2 (4D11), and NKG2D (CX5) (BD Biosciences), and Ly49I (YLI-90) and NKG2A (20d5) (eBioscience). The 4LO3311 (Ly49C) hybridoma was a kind gift from Suzanne Lemieux. All surface staining was performed in phosphate-buffered saline (PBS) after blocking unspecific staining via FcγRII/III with purified anti-CD16/32 (2.4G2) (MabTech). Dead cells were excluded from the gating with the Aqua dead cell dye (Life Technologies). Flow cytometric analyses were performed on an LSRII or FACSCalibur (Becton Dickinson) and data analysis was performed using the FlowJo software (Tree Star). For comparison of relative expression levels of specific NK-cell subsets, we have normalized the expression data (MFI) of all NK-cell subsets in every individual mouse to the mean of the MFI of the same NK subset in all $K^{b-/-}D^{b-/-}$ mice in the same experiment.

**Tumours.** RMA (H-2$^b$) is a subline derived in our laboratory from the B6 mouse-derived EL-4 T-cell leukemia[62], a kind gift from Prof G. Klein and RMA-S is a TAP2-deficient variant of RMA[63], also derived in our laboratory. Leukaemia cells were propagated as ascites lines in irradiated B6 mice. YAC-1 is a Moloney murine leukaemia virus induced T-cell lymphoma of the A/Sn strain which was maintained *in vitro*. YAC-1 and EL-4 cells were obtained from G Klein, Karolinska Institutet.

**Assessment of NK-cell activity *in vivo*.** *In vivo* cytotoxicity assays were performed as previously described[36]. Briefly, single-cell suspensions of spleen cells ($B2m^{-/-}$, B6) were labelled with a high (target cells: $B2m^{-/-}$) or low (control cells: B6) dose of CFSE (Invitrogen Molecular Probes). Target and control cells were mixed at a 1:1 ratio and co-injected intravenously (i.v.) $20 \times 10^6$ (spleen cells) cells. Spleens were harvested after 2 days and relative percentages of CFSE-positive cells in each population were measured by flow cytometry. Rejection is given as the relative survival of target cells = (% remaining control cells/% control cells in inoculate)/(% remaining targets/% targets in inoculate).

**In vivo blockade.** *In vivo* blockade of DNAM-1 was performed as previously described[28]. Briefly, mice were initially injected i.v. with 400 µg anti-DNAM-1 (mAb 3B3). After this time point mice were repeatedly injected every 5 days with 200 µg of mAb. After 48 h, or 14 days, the *in vivo* capacity of NK cells to kill i.v. injected spleen cells and the maturation pattern of NK cells were assessed.

**In vitro cytotoxicity assay.** IL-2-stimulated NK cells were generated by culturing sorted (MACS Miltenyi) splenic NK cells in complete αMEM medium (αMEM, 10 mM Hepes, 20 μM 2-mercaptoethanol, 10% FBS, 100 U ml$^{-1}$ penicillin, 100 U ml$^{-1}$ streptomycin) for 5 days with 1,000 U rIL-2 (Immunotools), previously described[28]. Target cells were incubated for 1 h in the presence of Na$_2^{51}$CrO$_4$ (Cr; Amersham) and then washed 3 × in PBS and incubated with effector cells at indicated effector:target (E:T) ratios. After 4 h, cell culture supernatants were collected and analysed by a radiation counter (Wallac, PerkinElmer). Specific lysis was calculated as follows: %specific lysis = ((experimental release − spontaneous release)/(maximum release − spontaneous release)) × 100.

**In vitro crosslinking assay.** To measure degranulation and IFN-γ production upon NK1.1 or NKp46 crosslinking, U-bottom plates were coated with 20 μg of anti-NK1.1 or anti-NKp46 for 1 h at 37 °C before being left overnight at 4 °C. The wells were then washed and 5 × 10$^4$ sorted NK cells (NK-cell isolation kit II, Miltenyi Biotec, cat.no: 130-096-892) were added to the wells. CD107a was added to the cultures for the duration of the assay (1:200). After 1 h, brefeldin A and monensin (Biolegend) were added to culture and incubated for another 3 h. Cells were collected and stained for surface markers and then fixed and permeabilized (BDCytofix/Cytoperm, BD Bioscience, cat. no: 554714). Intracellular IFN-γ was detected by staining with anti-IFN-γ antibody (Biolegend).

**BM preparation and surface staining.** BM was harvested from donor mice by crushing bones and removing debris using pre-separation filters (Milteny Biotech). Unfractionated BM cells (5 × 10$^6$ per 100 μl) were surface stained as indicated in Supplementary Fig. 8. The cells positive for the following markers: CD11b, Gr-1, Ter119, CD19, NK1.1, CD11c and CD3 were considered as lineage positive and excluded from further analysis. The source of all antibodies used is described above.

**NK-cell preparation and adoptive transfer.** Splenic NK cells were isolated by magnetic sorting with the NK-cell isolation kit II (Miltenyi Biotec) according to the manufacturer's instructions. The purity of the isolate was assessed by Flow Cytometry. In total, 1–3 × 10$^6$ NK cells were injected i.v. to irradiated (8 Gy) mice.

**Statistics and multivariate analyses.** Statistical analyses (except for Fig. 4) were conducted using GraphPad Prism 5. Either non-paired or paired two-tailed Students $t$-test and one-way analysis of variance (ANOVA; for frequencies) or Mann–Whitney $U$-test and repeated measures ANOVA (for MFI) were performed for data from at least three independent experiments, unless indicated otherwise. Error bars denote s.d. $P$ values are depicted as: *$P < 0.05$, **$P < 0.01$, ***$P < 0.001$.

For multivariate analyses (Fig. 4 and Supplementary Fig. 5; see also Supplementary Note 1) SIMCA software version 14 (Umetrics) was used. PCA was used to investigate underlying trends in the data, while orthogonal projections to latent structures with discriminant analysis was used to investigate the difference between defined groups (mouse strains).

**Data availability.** The datasets generated during and/or analysed during the current study are available from the corresponding author on reasonable request.

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

## Acknowledgements

We thank Kenth Andersson and Helén Braxenholm for expert assistance with *in vivo* experiments, Birgitta Wester for flow cytometry help, Elina Staaf for introduction to and teaching of the multivariate analyses software and Maria H. Johansson for critical reading and comments on the manuscript. Members of Petter Höglund's, Klas Kärre's and Benedict Chamber's group are acknowledged for stimulating discussions. This work was supported by Swedish Cancer Society (B.J.C., K.K.), the Swedish Foundation for Strategic Research (B.J.C., K.K.), Stockholm County Council Theme Center Grant (B.J.C.), the Swedish Childhood Cancer Foundation (K.K.) and the Swedish Research Council (K.K.).

## Authors contributions

Conception and design by A.K.W., N.K., P.H., K.K., B.J.C. Development of methodology by A.K.W., N.K., G.B., M.C., S.G., B.J.C. Acquisition of data by A.K.W., N.K., J.S. Analysis and interpretation of data by A.K.W., N.K., K.K., B.J.C. Manuscript was written by A.K.W., K.K. and B.J.C.

## Additional information

**Competing interests:** The authors declare no competing financial interests.

