## [Peer Review File · Nature Communications]

Reviewers' comments:

Reviewer #1 (Remarks to the Author):

NK cell education is thought to be an important process through which NK cells calibrate their function to the microenvironment and MHC class I molecules appear to be the major regulators of the process through interactions with inhibitory receptors on NK cells. The signals regulating this process however are not clear.

Wagner et al perform a thorough analysis of DNAM-1 expression in a number of mutant mouse strains that express defined patterns of specific MHC class I molecules, including strains deficient for MHC molecules. The conclusion is that DNAM-1 associates with but is not required for NK cell education.

I would suggest however the authors clarify for the reader the major advance over and beyond previously published papers (i.e. ref 25). More broadly, the authors may make an effort to compare mouse and human biology, for example DNAM-1 is expressed on all NK cells, differently from the mouse.

The merit of this paper is the unprecedented analysis, which is carefully done. The PCA is an example of how one can go about dissecting the complex diversity of NK cell repertoires. What do the authors mean by 77.2% 'predictability' in Fig legend 4?

The finding that DNAM-1 is expressed early in NK cell ontogeny is novel and perhaps merits to be emphasised more?

Figure 6. A typical such assay is done overnight. Here 44hours. Any specific reason why?

Figure 9. It would be informative to clarify the signals that may regulate the association of DNAM-1 with education.

The discussion is balanced and interesting but perhaps the authors should commit more to propose how they think it works. What are the signals involved that connect education with DNAM-1? Or is the association only correlative?

How does DNAM-1 and education impact the generation of NK cell memory? DNAM-1 seems necessary for MCMV-induced Ly49H+ expansion (Nabekura Immunity 2014).

The 'tunable' association DNAM-1 with Ly49 compared with the stable association with NKG2A deserves perhaps to be put in context. For example, how does it relate to the recent finding in human NK cells from the Stanford group that NK cell education is imparted by two 'schools' of KIR and NKG2A education? (Science Immunol 2016).

Reviewer #2 (Remarks to the Author):

In their manuscript titled 'Expression of DNAM-1 (CD226) is associated to but not required for NK cell education', Wagner and co-workers have explored the link between DNAM-1 expression and NK cell education. The methodological approaches are extensive and include phenotypic characterization, expression analysis, in vitro and in vivo analysis of NK cell functions, and adoptive transfer into MHC-

I-deficient mice. They have extensively characterized DNAM-1 expression on different NK cell subsets based on the expression of inhibitory self-MHC-I receptors and demonstrate an association between DNAM-1 expression and NK cell education. DNAM-1 expression was found to be modulated in MHC-I-dependent manner. They also show a link between DNAM-1 and NKG2A expression which is independent of MHC-I. Killing of MHC-I-deficient target cells by MHC-I-educated NK cells was dependent on DNAM-1 expression but missing-self responses were largely intact in DNAM-1^{-/-} mice. The authors conclude that DNAM-1 expression is required for educated NK cell's effector functions but is dispensable for NK cell education.

Various models have been put forward but none have so far completely described the process of NK cell education. This study is interesting and a step forward in understanding this complex and poorly understood phenomenon in NK cell biology. However, few queries need to be addressed.

Major queries:

1. The manuscript does not provide enough background on DNAM-1 expression and function. Better introduction with more background on DNAM-1 is necessary to put this study in the context of the existing literature.
2. In Figure 2b, higher DNAM-1 expression is observed on Ly49I⁺ NK cells in KbDb mice but not in Kb-single, Db-single and KbDbDd mice. Authors have not commented on inconsistencies in the correlation between DNAM-1 expression and inhibitory Ly49I receptor.
3. In Figure 3, authors correlate DNAM-1 expression to the number of inhibitory Ly49 receptors. However, this figure treats all Ly49 receptors in B6 mice equally in imparting NK cell education. Authors may consider color-coding data points based on the Ly49 receptor type for better interpretation of data.
4. Figure 4 in general lacks a good description and is difficult to understand for those who are not familiar with this type of analysis. Moreover, it recapitulates the data shown in previous figures. This figure can be included in the supplemental data and better description is needed.
5. Were freshly isolated NK cells used in the killing assays shown in Figure 5? Does DNAM-1^{-/-} NK cells kill MHC-I deficient target cells?
6. Is DNAM-1 also required for IFN γ production, degranulation (CD107a expression) and granzyme B expression by educated NK cells upon in vitro stimulation? This data should be included in the manuscript.
7. Authors show no difference in in vivo killing of MHC-I-deficient cell in B6 mice and those treated with anti-DNAM-1 antibody in Figure 6c. However, only 40% of MHC-I-deficient cells are killed in B6 mice compared to the 75% killing shown in Figure 6a. Do authors consistently see no difference in the killing of MHC-I-deficient cell in anti-DNAM-1-treated and untreated B6 mice in independent repeats of this experiment?
8. Authors only describe DNAM-1 association with MHC-I-mediated education via Ly49 receptors. Is DNAM-1 also associated with NK cell education via NKG2A receptor?
9. Have they also looked at the association between DNAM-1 expression and NK cell education in DNAM-1 ligand-deficient mice?
10. DNAM-1⁺ NK cell frequency was shown to be lower in presences of the activating Ly49D receptor and H-2Dd ligand. How is DNAM-1 expression and frequency affected by other activating receptors, such as NKG2C?
11. The discussion is weak and highly speculative. A correlation between this study and previous human studies is missing. Authors must discuss their finding in the context of previous human studies.

Minor queries:

1. The use of abbreviations such as Ly49r, Ly49R and Ly49rs for Ly49 receptors throughout the manuscript and figures is confusing. Authors must refrain from using such abbreviations.
2. Several typographical errors are apparent throughout the manuscript.

Reviewer #3 (Remarks to the Author):

The study by Wagner and collaborators aims to evaluate a role of DNAM-1 in NK cell education and to demonstrate a possible correlation between expression of the MHC I-specific Ly49 receptors and % of DNAM-1 expressing NK cells and/or DNAM-1 surface density. To this end they performed an extensive phenotypic characterization of DNAM-1+ cells using different mouse models.

The results include some interesting preliminary data, however they are generally poorly organized, confusing and sometimes contradictory. Overall, data demonstrate the importance in both activating receptor expression and NK cell function of co-expressing an inhibitory MHC-I receptor. This is not new. Moreover, results don't clearly demonstrate the requirement of either educated or non-educated Ly49 receptors.

The authors show a very early expression of DNAM-1 during NK cell development. However, they don't analyze its function or take into consideration the existence of an inhibitory counterpart of DNAM-1 that could play a role during the process. Moreover, it is quite surprising that NKp46 expression wasn't evaluated in comparison to DNAM-1 expression, particularly during maturational stages.

REVIEWERS' COMMENTS:

Reviewer #1 (Remarks to the Author):

The Authors have adequately addressed most of my concerns

Reviewer #2 (Remarks to the Author):

I am satisfied with the revisions and responses to my previous comments. I congratulate the authors for this important work.

Reply To Reviewer 1

Point 1 I would suggest however the authors clarify for the reader the major advance over and beyond previously published papers (i.e. ref 25). More broadly, the authors may make an effort to compare mouse and human biology, for example DNAM-1 is expressed on all NK cells, differently from the mouse.

Author reply

In the discussion, we have stated more explicitly how our data extends previously published literature, including how this leads us to the model that we propose. In the introduction and in parts of the discussion, we have brought up the differences between the expression of DNAM-1 on mouse and human NK cells. We bring up the point that in young mice nearly all NK cells express DNAM-1 and that in aging humans and in some diseases in humans, NK cells lose DNAM-1 expression. While the expression patterns are different in adult mice this may be more of a reflection of mice living in controlled environments rather than a definitive differences between species.

Point 2 The merit of this paper is the unprecedented analysis, which is carefully done. The PCA is an example of how one can go about dissecting the complex diversity of NK cell repertoires. What do the authors mean by 77.2% 'predictability' in Fig legend 4?

Author reply

We thank the reviewer for bringing this to our attention. The “77.2% predictability” in the legend for Figure 4 stems from a previous version of the figure, where the OPLS-DA (Suppl. Fig. 4) was still included in the main manuscript. In an OPLS-DA, a test set (a randomly chosen part of the entire dataset) is used to estimate the predictability of another set (a different randomly chosen part of the dataset). This is done in an iterative way, so that every possible division of the entire dataset is used for the estimate of predictability. A high predictability is of course a measurement of quality of the model. Fig 4 as it is now, does only show the PCA of the dataset, an algorithm which can reduce the dimensionality of a given dataset, where the first principal component (PC) accounts for most of the variability of the dataset. Each succeeding component shows the highest possible variance after exclusion of previous PCs. We have therefore updated the figure legend for Fig. 4, and have explained the predictability of the OPLS-DA in more detail in the Supplemental Material.

Point 3 The finding that DNAM-1 is expressed early in NK cell ontogeny is novel and perhaps merits to be emphasised more?

Author reply

We thank the reviewer for bringing up this point. We have now highlighted this point in the discussion.

Point 4 Figure 6. A typical such assay is done overnight. Here 44hours. Any specific reason why?

Author reply

While we agree with the reviewer that the in vivo rejection assay can be run overnight as we described in our original publication of this assay for NK cells, we actually found in this original publication that 44 hours was optimal for the assay when performed with spleen cells (Oberg, Johansson et al. 2004).

Point 5 Figure 9. It would be informative to clarify the signals that may regulate the association of DNAM-1 with education.

Author reply

In the discussion and Figure 10, we have brought up how signals from MHC class I molecules would control DNAM-1 expression in NK cells. We actually envisage two separate signals, one which is controlled by NKG2A expression that is independent of MHC class I expression and the other via MHC class I molecules and education. This suggests that inhibitory signals and molecules contributing to baseline activation thresholds could modulate DNAM-1 expression, which has been actually observed in mice lacking SHIP-1 (Gumbleton, Vivier et al. 2015).

Point 6 The discussion is balanced and interesting but perhaps the authors should commit more to propose how they think it works. What are the signals involved that connect education with DNAM-1? Or is the association only correlative?

Author reply

As stated in the paper, we conclude that DNAM1 expression is not a requirement for education but is correlated with educated NK cells. However, in our experiments with adoptive transfer of NK cells, regulation of DNAM-1 expression and education appear to be controlled by the same signals. We have expanded more in the discussion about how we believe DNAM-1 could be connected to education.

Point 7 How does DNAM-1 and education impact the generation of NK cell memory? DNAM-1 seems necessary for MCMV-induced Ly49H+ expansion (Nabekura Immunity 2014).

Author reply

This is an interesting question. In the scope of the present study we have not looked at “memory” NK cells in vivo. However one could postulate that the degree of education among the DNAM-1⁺ cells might allow the cells to be under more sustained control and thus would not induce so much activation-induced cell death.

There seems to be contradictory results published when it comes to expression of self-specific inhibitory receptors and the formation of memory NK cells. In the hapten-induced contact hypersensitivity model, education seems to be beneficial if not necessary (O'Leary, Goodarzi et al. 2006). Similarly in humans, NKG2C⁺ adaptive NK cells express KIRs specific for self-HLA-I (Beziat, Dalgard et al. 2012; Foley, Cooley et al. 2012; Foley, Cooley et al. 2012; Romee, Schneider et al. 2012; Beziat, Liu et al. 2013). The results from mouse studies

using hapten-induced memory, and the human studies indicate that NK cell education may be helpful in the generation of memory cells. However, the memory NK cells that are generated during MCMV infection are primarily negative for self-MHC-I-specific Ly49r, and it has been shown that it is rather the “unlicensed” NK cells that protect the mouse and help clear an infection of MCMV or influenza (Orr, Murphy et al. 2010; Mahmoud, Tu et al. 2016). Furthermore, DNAM-1 is required for expansion of Ly49H⁺ NK cells and formation of memory in the MCMV model (Nabekura, Kanaya et al. 2014). However in our previous publication (Seth, Georgoudaki et al. 2009), and in the current study, we saw that Ly49H expression is actually lower on the DNAM-1⁺ NK cells. We have found that the NKG2A⁺ DNAM-1⁺ NK cells do expression less Ly49H and so in terms of education and memory maybe the NKG2A⁻ DNAM1⁺ Ly49H⁺ NK cells might denote a specialized memory less educated NK cell subset. This is perhaps an area for further study as it might be more complex than the publication by Nabekura originally alluded to.

Obviously this discussion is too long for the manuscript however in response to the referee’s request, we have added short relevant comments within the discussion about this.

Point 8 The 'tunable' association DNAM-1 with Ly49 compared with the stable association with NKG2A deserves perhaps to be put in context. For example, how does it relate to the recent finding in human NK cells from the Stanford group that NK cell education is imparted by two 'schools' of KIR and NKG2A education? (Science Immunol 2016).

Author reply

We agree that it is relevant to bring up this interesting new concept. We have now covered this comparison in the discussion. However, the data from our paper are likely not relevant to the Stanford model, since DNAM-1 expression and NKG2A co-expression can occur even in the absence of MHC class I molecules. Interestingly though, retuning seems to be differently regulated for different receptors which we discuss.

Reply To Reviewer 2

Major queries:

Point 1. The manuscript does not provide enough background on DNAM-1 expression and function. Better introduction with more background on DNAM-1 is necessary to put this study in the context of the existing literature.

Author reply

We have expanded the introduction and discussion to include more of the current literature as well as differences/similarities between DNAM-1 expression on mouse and human NK cells

Point 2. In Figure 2b, higher DNAM-1 expression is observed on Ly49I+ NK cells in KbDb mice but not in Kb-single, Db-single and KbDbDd mice. Authors have not commented on inconsistencies in the correlation between DNAM-1 expression and inhibitory Ly49I receptor.

Author reply

We thank the reviewer for picking this up as it is an interesting observation. Ly49I binds to K^b in most mouse strains. The functional consequence of this interaction, is however different depending on which outcome is measured and which assay and mouse strain are used (e.g. cytokine production, killing, degranulation, receptor down-modulation, tetramer binding). While Ly49I single-positive NK cells are clearly educated in the presence of K^b (Hanke, Takizawa et al. 1999; Michaelsson, Achour et al. 2000; Wagner, Wickstrom et al. 2016), the inhibitory receptor is not downregulated, while Ly49C, which also recognizes and is educated by K^b, exhibits down-modulation in the presence of the ligand (Johansson, Johansson et al. 2005). One difference is that Ly49C can also bind to K^b in cis, while this has not been demonstrated for Ly49I. Furthermore, in our previous publication on retuning of NK cell education, we show that Ly49I single-positive cells are re-educated while Ly49C-single-positive cells do not change their responsiveness after blockade of the inhibitory receptors (Wagner, Wickstrom et al. 2016).

Our data in this study does not allow a clear-cut conclusion on the influence of Ly49I on DNAM-1 levels in terms of MFI. However, when looking at the frequency of NK cells that are positive for DNAM-1 (Suppl. Fig. 2b) one can see the influence of the K^b-Ly49I interaction not only in B6 (K^bD^b) but also in Kb-single mice. This is stated in the results section. That this effect is not visible in D8 mice (K^bD^bD^d), most likely due to the fact that the interaction of Ly49A with its ligand D^d is dominant (Brodin, Lakshmikanth et al. 2009).

Point 3. In Figure 3, authors correlate DNAM-1 expression to the number of inhibitory Ly49 receptors. However, this figure treats all Ly49 receptors in B6 mice equally in imparting NK cell education. Authors may consider color-coding data points based on the Ly49 receptor type for better interpretation of data.

Author reply

It is correct that in Fig.3, all inhibitory Ly49r are treated equally. We have shown in the previous Figure (Fig. 2 and Suppl Fig. 2) that specific Ly49r together with the corresponding MHC-I allele have an impact on DNAM-1 expression. In this figure, we want to make the point that there are 2 superimposed mechanisms that influence DNAM-1 expression, the first being expression of Ly49r, regardless of specificity of the Ly49r, and the second being the tight correlation to NKG2A. The way this is calculated does not allow color-coding with respect to Ly49r type. Every dot represents the data of all NK cell subsets with the same number of inhibitory Ly49r in one mouse. So for every mouse, there is one dot for 0 receptors, 1 receptor, 2 receptor and so forth.

We realize that we could have been more precise in the previously submitted version, and we have now clarified this in the Figure legend.

Point 4. Figure 4 in general lacks a good description and is difficult to understand for those who are not familiar with this type of analysis. Moreover, it recapitulates the data shown in previous figures. This figure can be included in the supplemental data and better description is needed.

Author reply

Since we believe that this is a good way to model our data in an unbiased way, as supported by reviewer 1, we have decided to keep the figure. However as we realise from reviewer 2, we could make the figure more didactic. We have reduced the amount of data in the figure to make it clearer and focused only on the mice lacking MHC class I molecules or expressing single MHC class I molecules. We have added to the text in the main body of the manuscript hopefully a more clarified explanation of how the PCA works.

Point 5. Were freshly isolated NK cells used in the killing assays shown in Figure 5? Does DNAM-1^{-/-} NK cells kill MHC-I deficient target cells?

Author reply

We used IL-2 stimulated NK cells in these killing assays (we have made this clearer in the figure legends and Material and Methods).

Yes, the IL-2 stimulated DNAM-1^{-/-} NK cells do kill MHC-I deficient target cells and this has been included in Figure 5. What is interesting with this finding is that it suggests in the absence of DNAM-1 there is still education, as the NK cells of DNAM-1^{-/-} mice can differentiate between RMA and RMA-S.

Point 6. Is DNAM-1 also required for IFN γ production, degranulation (CD107a expression) and granzyme B expression by educated NK cells upon in vitro stimulation? This data should be included in the manuscript.

Author reply

Previous publications have shown that DNAM1⁺ NK cells of the total NK cell population readily produce IFN γ upon IL-12 and IL18 stimulation, although DNAM-1⁺ NK cells produce significantly more IFN γ . We have now included additional data showing degranulation and IFN γ production specifically in MHC class I-educated NK cell subsets upon crosslinking of activating receptors (Figure 6). These data clearly show that DNAM-1 is not required for IFN γ production or CD107 expression. This is consistent with the model proposed in the discussion (Figure 10).

Point 7. Authors show no difference in in vivo killing of MHC-I-deficient cell in B6 mice and those treated with anti-DNAM-1 antibody in Figure 6c. However, only 40% of MHC-I-deficient cells are killed in B6 mice compared to the 75% killing shown in Figure 6a. Do authors consistently see no difference in the killing of MHC-I-deficient cell in anti-DNAM-1-treated and untreated B6 mice in independent repeats of this experiment?

Author reply

Regarding the point raised by the reviewer, the measurement of missing self response is quite reproducible within each set of experiments but can show some quantitative variation between sets of experiments. The reason

for this is unclear but may have to do with conditions in the mouse house etc. This is why in the new Fig. 7 the survival ratio for MHC-I- spleen cells is 0.3 in the first set of experiments (Fig. 7a) and 0.55 in the second set of experiments (Fig. 7c).

The essential point is that one has a control to assure that there is a missing self response by using appropriate negative controls, such as $\beta 2m^{-/-}$ or MHC-I^{-/-} mice as recipients. We have included these mice in every experiment but did not show this control in the previously submitted version. We have now included the $\beta 2m$ control in the new Fig 7c.

We realize that we did not specify how many mice and how many experiments were performed for Fig. 7 (Figure 6 in the previously submitted version). This has been clarified now in the Figure legend.

Point 8. Authors only describe DNAM-1 association with MHC-I-mediated education via Ly49 receptors. Is DNAM-1 also associated with NK cell education via NKG2A receptor?

Author reply

We have now made this more clear in the discussion that there is a connection between NKG2A and DNAM-1 expression but this is independent of MHC class I expression in the host.

Point 9. Have they also looked at the association between DNAM-1 expression and NK cell education in DNAM-1 ligand-deficient mice?

Author reply

Yes we have. This is an ongoing study but below we provide preliminary data for the reviewer. DNAM-1 expression is higher on CD155^{-/-} cells, as previously published (Seth, Qiu et al. 2011), both in terms of frequency and expression level per cell (CD155 Figure, A, B, C). This increase of DNAM-1 levels is seen on every NK cell subset, independently of NKG2A or Ly49r co-expression (not shown). The frequencies of Ly49r-single positive NK cell subsets change slightly, as shown for Ly49I-sp and NKG2A-sp (CD155 Figure D). Missing self rejection of $\beta 2m$ -negative target cells in vivo is comparable in CD155^{-/-} and B6 wt mice (CD155 Figure E).

NK cells in CD155^{-/-} mice

CD155 Figure

(A-C) Expression DNAM-1 is higher on CD155^{-/-} NK cells. (A) histogram of DNAM-1 expression on CD155^{-/-} and B6 NK cells, (B, C) compiled data of 6 mice per group of 3 independent experiments. (D) Frequency of selected NK cell subsets based on their expression of Ly49r and NKG2A. Gated on live CD3⁺NK1.1⁺ cells in B6 and CD155^{-/-} mice. Data is compiled from 6 mice of 3 independent experiments. (E) In vivo rejection assay of β 2m^{-/-} spleen cells 44h after inoculation. 2 independent experiment with 6 mice per group.

Point 10. DNAM-1+ NK cell frequency was shown to be lower in presences of the activating Ly49D receptor and H-2Dd ligand. How is DNAM-1 expression and frequency affected by other activating receptors, such as NKG2C?

Author reply

A supplementary figure on activating receptors has now been included (Suppl. Fig. 4). As previously published, both Ly49D and Ly49H were less frequent on DNAM-1⁺ cells (Seth, Georgoudaki et al. 2009), while other activating receptors such as NK1.1, NKG2D, Nkp46 and 2B4 showed a higher expression of DNAM-1⁺ NK cells compared to DNAM-1⁻ NK cells within B6 mice (Supplemental Fig. 4e), or on total NK cells from B6 mice compared to DNAM-1^{-/-} mice (Supplemental Fig. 4f).

Point 11. The discussion is weak and highly speculative. A correlation between this study and previous human studies is missing. Authors must discuss their finding in the context of previous human studies.

Author reply

We have tried to clarify conclusions versus speculations in the revised discussion. We have also highlighted in the discussion more about the differences and similarities in DNAM-1 expression on human and mouse NK cells. Furthermore, we also discuss our model for murine NK cells in the context of human NK cell biology.

Minor queries:

Points 1. The use of abbreviations such as Ly49r, Ly49R and Ly49rs for Ly49 receptors throughout the manuscript and figures is confusing. Authors must refrain from using such abbreviations.

Author reply

We have now been more consistent with our use of these abbreviations.

Point 2. Several typographical errors are apparent throughout the manuscript.

Author reply

Hopefully these have been found and corrected.

Reply To Reviewer 3

The study by Wagner and collaborators aims to evaluate a role of DNAM-1 in NK cell education and to demonstrate a possible correlation between expression of the MHC I-specific Ly49 receptors and % of DNAM-1 expressing NK cells and/or DNAM-1 surface density. To this end they performed an extensive phenotypic characterization of DNAM-1+ cells using different mouse models.

The results include some interesting preliminary data, however they are generally poorly organized, confusing and sometimes contradictory. Overall, data demonstrate the importance in both activating receptor expression and NK cell function of co-expressing an inhibitory MHC-I receptor. This is not new. Moreover, results don't clearly demonstrate the requirement of either educated or non-educated Ly49 receptors.

The authors show a very early expression of DNAM-1 during NK cell development. However, they don't analyze its function or take into consideration the existence of an inhibitory counterpart of DNAM-1 that could play a role during the process. Moreover, it is quite surprising that Nkp46 expression wasn't evaluated in comparison to DNAM-1 expression, particularly during maturational stages.

Author reply

We disagree with the reviewer that there is nothing new in this paper and would like to highlight these points. Firstly, DNAM-1 is connected to NKG2A independently of MHC class I. Secondly even though there is a close relationship between DNAM-1 and education as others have previously published, we can show based on in vivo and in vitro experiments that DNAM-1 is not needed for education or maintaining education. Thirdly DNAM-1 is expressed early in the ontogeny of NK cells and is gradually lost with differentiation, maturation and age. Moreover, our data are the first to show that DNAM-1 expression is plastic depending on the MHC class I environment the NK cells find themselves in. Finally we propose a model that represents a considerable extension of the one discussed recently in the literature (Martinet, Ferrari De Andrade et al. 2015).

Based on the reviewer's suggestion, we have now assessed TIGIT expression on NK cells in relation to education. We have found that in NK cells, TIGIT exhibits an expression pattern opposite to that of DNAM-1, i.e. higher in MHC-I^{-/-} mice and higher on NK cell subsets that don't bind to the hosts MHC alleles. This has now been included in the results and in the supplemental data (Suppl. Fig. 3). The other counter-receptor, CD96, is expressed on all mature NK cells, and correspondingly, we have not seen any role for CD96 in education. CD96 is expressed early on NK cell precursors in the BM, and TIGIT expression is only measurable after DNAM-1 and NKG2A expression on NK1.1⁺ NK cells. This data is currently being prepared for another manuscript.

Expression of DNAM-1 occurs before Nkp46 expression during NK cell development. We did not assess Nkp46 expression in relation to NK cell development, as it was shown that the precursors that can be identified by this staining and gating strategy we have employed do not express Nkp46, but give rise to mature Nkp46⁺ NK cells after transplantation into mice (Fathman, Bhattacharya et al. 2011).

Literature:

- Beziat, V., O. Dalgard, et al. (2012). "CMV drives clonal expansion of NKG2C+ NK cells expressing self-specific KIRs in chronic hepatitis patients." *Eur J Immunol* **42**(2): 447-457.
- Beziat, V., L. L. Liu, et al. (2013). "NK cell responses to cytomegalovirus infection lead to stable imprints in the human KIR repertoire and involve activating KIRs." *Blood* **121**(14): 2678-2688.
- Brodin, P., T. Lakshmikanth, et al. (2009). "The strength of inhibitory input during education quantitatively tunes the functional responsiveness of individual natural killer cells." *Blood* **113**(11): 2434-2441.
- Fathman, J. W., D. Bhattacharya, et al. (2011). "Identification of the earliest natural killer cell-committed progenitor in murine bone marrow." *Blood* **118**(20): 5439-5447.
- Foley, B., S. Cooley, et al. (2012). "Human cytomegalovirus (CMV)-induced memory-like NKG2C(+) NK cells are transplantable and expand in vivo in response to recipient CMV antigen." *J Immunol* **189**(10): 5082-5088.
- Foley, B., S. Cooley, et al. (2012). "Cytomegalovirus reactivation after allogeneic transplantation promotes a lasting increase in educated NKG2C+ natural killer cells with potent function." *Blood* **119**(11): 2665-2674.
- Gumbleton, M., E. Vivier, et al. (2015). "SHIP1 intrinsically regulates NK cell signaling and education, resulting in tolerance of an MHC class I-mismatched bone marrow graft in mice." *J Immunol* **194**(6): 2847-2854.
- Hanke, T., H. Takizawa, et al. (1999). "Direct assessment of MHC class I binding by seven Ly49 inhibitory NK cell receptors." *Immunity* **11**(1): 67-77.
- Johansson, S., M. Johansson, et al. (2005). "Natural killer cell education in mice with single or multiple major histocompatibility complex class I molecules." *J Exp Med* **201**(7): 1145-1155.
- Mahmoud, A. B., M. M. Tu, et al. (2016). "Influenza Virus Targets Class I MHC-Educated NK Cells for Immuno-evasion." *PLoS Pathog* **12**(2): e1005446.
- Martinet, L., L. Ferrari De Andrade, et al. (2015). "DNAM-1 expression marks an alternative program of NK cell maturation." *Cell Rep* **11**(1): 85-97.
- Michaelsson, J., A. Achour, et al. (2000). "Visualization of inhibitory Ly49 receptor specificity with soluble major histocompatibility complex class I tetramers." *Eur J Immunol* **30**(1): 300-307.
- Nabekura, T., M. Kanaya, et al. (2014). "Costimulatory Molecule DNAM-1 Is Essential for Optimal Differentiation of Memory Natural Killer Cells during Mouse Cytomegalovirus Infection." *Immunity*.
- O'Leary, J. G., M. Goodarzi, et al. (2006). "T cell- and B cell-independent adaptive immunity mediated by natural killer cells." *Nat Immunol* **7**(5): 507-516.
- Oberg, L., S. Johansson, et al. (2004). "Loss or mismatch of MHC class I is sufficient to trigger NK cell-mediated rejection of resting lymphocytes in vivo - role of KARAP/DAP12-dependent and -independent pathways." *Eur J Immunol* **34**(6): 1646-1653.
- Orr, M. T., W. J. Murphy, et al. (2010). "'Unlicensed' natural killer cells dominate the response to cytomegalovirus infection." *Nat Immunol* **11**(4): 321-327.
- Romee, R., S. E. Schneider, et al. (2012). "Cytokine activation induces human memory-like NK cells." *Blood* **120**(24): 4751-4760.
- Seth, S., A. M. Georgoudaki, et al. (2009). "Heterogeneous expression of the adhesion receptor CD226 on murine NK and T cells and its function in NK-mediated killing of immature dendritic cells." *J Leukoc Biol* **86**(1): 91-101.
- Seth, S., Q. Qiu, et al. (2011). "Intranodal interaction with dendritic cells dynamically regulates surface expression of the co-stimulatory receptor CD226 protein on murine T cells." *J Biol Chem* **286**(45): 39153-39163.

Wagner, A. K., S. L. Wickstrom, et al. (2016). "Retuning of Mouse NK Cells after Interference with MHC Class I Sensing Adjusts Self-Tolerance but Preserves Anticancer Response." Cancer Immunol Res 4(2): 113-123.